# Power laws in species' biotic interaction networks can be inferred from co-occurrence data

Nuria Galiana [1,5] ✉, Jean-François Arnoldi[2,5], Frederico Mestre[3], Alejandro Rozenfeld[3,4] & Miguel B. Araújo [1,3]

Inferring biotic interactions from species co-occurrence patterns has long intrigued ecologists. Yet recent research revealed that co-occurrences may not reliably represent pairwise biotic interactions. We propose that examining network-level co-occurrence patterns can provide valuable insights into community structure and assembly. Analysing ten bipartite networks of empirically sampled biotic interactions and associated species spatial distribution, we find that approximately 20% of co-occurrences correspond to actual interactions. Moreover, the degree distribution shifts from exponential in co-occurrence networks to power laws in networks of biotic interactions. This shift results from a strong interplay between species' biotic (their interacting partners) and abiotic (their environmental requirements) niches, and is accurately predicted by considering co-occurrence frequencies. Our work offers a mechanistic understanding of the assembly of ecological communities and suggests simple ways to infer fundamental biotic interaction network characteristics from co-occurrence data.

Whether interactions between species can be inferred from species co-occurrence patterns is a contentious topic in ecology[1–4]. Indeed, the spatial distribution of species is influenced by their environmental tolerances and dispersal abilities, but also by the interactions they establish with other species[5–8]. This has led to a century-long debate, with some authors arguing that biotic interactions should leave detectable patterns in species co-occurrences[9–12]. Consequently, there have been expectations that biotic interactions could be inferred, at least to some extent, from co-occurrence data. However, this straightforward approach has faced criticism because of the potential blurring of signals by other factors that constrain species ranges and coexistence[13,14]. For instance, species that do not interact but share physiological or habitat requirements might lead to false inferences of biotic interactions. Conversely, negative interactions resulting in avoidance or exclusion may generate non-overlapping distributions, making it difficult to identify a clear signal in co-occurrence data[2,13,14].

Empirically testing the predictive capacity of co-occurrences as a surrogate of biotic interactions presents important challenges. Few studies that have explored this relationship have managed to establish general rules for inferring connections between these two expressions of species ecologies. For example, Freilich et al.[15] discovered a weak correspondence between interactions inferred from co-occurrences and the actual biotic interactions observed in a rocky intertidal in central Chile. Only approximately half of the known interactions were

[1]Department of Biogeography and Global Change, National Museum of Natural Sciences, Madrid, Spain. [2]Centre National de la Recherche Scientifique, Experimental and Theoretical Ecology Station, Moulis, France. [3]Rui Nabeiro Biodiversity Chair, Mediterranean Institute for Agriculture, Environment and Development, University of Évora, Évora, Portugal. [4]INTELYMEC Group, Centro de Investigaciones en Física e Ingeniería del Centro Centro de Investigaciones en Física e Ingeniería del Centro de la Provincia de Buenos Aires – Universidad Nacional del Centro de la Provincia de Buenos Aires – Consejo Nacional de Investigaciones Científicas y Técnicas, Olavarría, Argentina. [5]These authors contributed equally: Nuria Galiana, Jean-François Arnoldi. ✉e-mail: galiana.nuria@gmail.com

accurately detected through co-occurrence data. Many interactions, especially negative interactions, were missed entirely, while many others were inaccurately classified as false or spurious interactions.

Although specific interactions may not be reliably inferred from co-occurrence data, there are still valuable insights to be gained by exploring other aspects of co-occurrence information to better understand how ecological networks come together. For instance, by establishing links between species that co-occur in space, we can analyse the structural properties of these co-occurrence networks[2,16] and investigate their connections with the structural properties of actual interaction networks (Fig. 1a). Complex ecological systems often exhibit well-defined patterns that can shed light on the underlying mechanisms[17-19]. In this vein, we propose that comparing the structures of co-occurrence and interaction networks can provide valuable information about the drivers of community assembly, circumventing the challenges associated with inferring specific pairwise interactions.

We focus on a fundamental aspect of network structure extensively studied in ecological communities: their 'degree distribution'[17,20]. The degree distribution encodes the probability for a randomly chosen node of the network to possess a certain number of links. In ecology, this concept is crucial in describing how links are distributed between species, and its shape can be related to specific aspects of ecological stability[17,18,20,21] (Fig. 1b and Supplementary Text 2). Scale-free (power-law) degree distributions are considered a hallmark of network organization, conferring robustness to random extinctions because highly connected species are relatively rare, making their removal less likely[22,23]. On the other hand, networks with an exponential degree distribution, which occurs when connections are established randomly, have many species with moderate connectivity, making the network more vulnerable to random species losses. The degree distribution of biotic interaction networks is thus a fundamental feature of ecological networks, contributing to the understanding of community structure and its response to environmental changes and other threats to persistence[24]. We propose that a deeper understanding can be achieved by combining the study of the degree distribution of co-occurrence networks because both types of networks may emerge from shared community assembly rules.

The simple premise on which the comparison between co-occurrence and interaction networks is based is that species need to co-occur to interact. While competitive exclusion leading to spatial avoidance is an exception, for many important interaction types (predator–prey, host–parasitoid, commensalism or mutualism), co-occurrence is a necessary condition for the interaction[25]. Formally, this implies that interaction networks are a subset of co-occurrence networks, resulting from the removal of links between species that co-occur without interacting. Under the assumption that co-occurrence is observed at a relevant scale (if the scale considered is too broad, co-occurrence is indeed meaningless), this process of pruning could leave subtle traces of community assembly. For instance, if interaction networks were the result of randomly removing co-occurrence links, it would suggest that interactions are mostly contingent, and the structure of interaction networks would be trivially subordinate to the co-occurrence network. On the other hand, if the structure of the interaction network differs qualitatively from the structure of the co-occurrence network (beyond a reduction of the total number of links), it is a sign that there is a systematic pruning of links. For instance, a change in the network degree distribution from exponential (co-occurrence) to power-law (biotic interaction) signals chance-based co-occurrence patterns but a biased pruning of links benefiting generalist species (see below and Supplementary Text 2). The comparison between co-occurrence networks and biotic interaction networks thus offers a unique opportunity to investigate the interplay between the abiotic niche of a species (that sets the limits to its spatial distribution) and its biotic niche (its interacting partners, such as its prey or predators). Understanding such an interplay can then elucidate key mechanisms of community assembly.

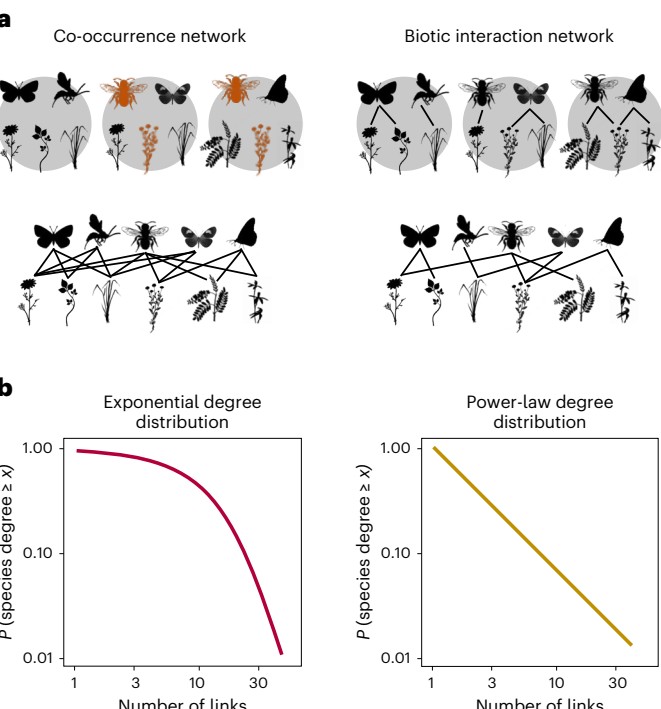

**Fig. 1 | Visual representation of a network of co-occurrence and the corresponding network of biotic interactions and their degree distribution.** **a**, A co-occurrence network is built based on empirical observations of presence and absence of each species in each spatial unit of each dataset. An interaction is added to the network of co-occurrences when two species from different trophic levels (for example, plant–pollinator or host–parasitoid) co-occur in at least one spatial unit. For instance, a pollinator will have as many interactions in the network of co-occurrences as plants found co-occurring with it in the spatial units analysed. Notice that the frequency of co-occurrence (that is, the number of times two species co-occur across spatial units) varies across species. Species coloured in orange have a higher frequency of co-occurrence. The network of biotic interactions is based on empirical observations of the interactions between species in each spatial unit. Thus, an interaction between two species is added to the network of biotic interactions if it was empirically observed in at least one spatial unit. **b**, Once both types of networks are built, we can analyse their fundamental characteristics. One of them is network degree distribution, which represents the cumulative probability of finding a species in the network with at least a given number of interactions. Therefore, the probability of finding a species in the network that has at least one interaction with another species is 1. The shape of the network degree distribution indicates how links are distributed among species in the network. For instance, an exponential shape indicates that the occurrence of a link in the network is independent of the presence of other links, while power-law distributions indicate that links are more likely to occur among species that already have more links (that is, the rich-gets-richer phenomenon).

To explore these concepts, we analysed ten well-resolved empirical bipartite networks from various terrestrial habitats around the globe. These networks provide data on both species distributions across multiple locations and their empirically sampled biotic interactions. Importantly, the interactions in these networks, such as plant–pollinator (PP) and host–parasitoid (HP) interactions, adhere to the premise that 'species need to co-occur (at an appropriate scale; see 'Results and discussion') to interact'. These newly available datasets allow us to directly compare co-occurrence and interaction networks in terms of their degree distribution (Fig. 1). We then examined how co-occurrence networks can be pruned to approximate the features of the realized biotic interaction network. We propose a simple model that uses knowledge on co-occurrence between species and the frequency of such co-occurrence (that is, the number of sites where co-occurrence

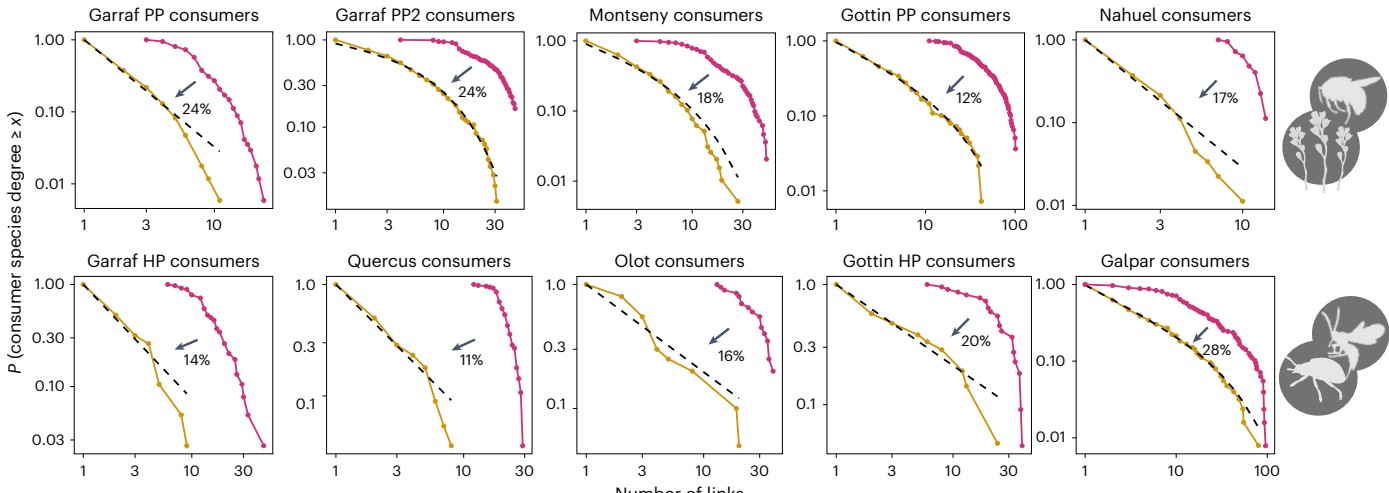

**Fig. 2 | Comparison of the frequency distributions of consumer's degree among co-occurrence networks and realized biotic interactions networks.** The red lines correspond to the degree distributions of the co-occurrence networks and the yellow lines correspond to the degree distribution of the biotic interaction networks in the ten datasets investigated. Top, PP networks. Bottom, HP networks. The black dashed lines indicate the power law or truncated power-law fit, which for all datasets were the more parsimonious functions among all tested (Methods). The percentage shown in each plot represents the proportion of links realized in the network of biotic interactions from the co-occurrence network (*f*) for each dataset. See the corresponding results for resource species in Extended Data Fig. 1. A description of all datasets can be found in Supplementary Text 1.

between a pair of species is known). This model estimates the number of interactions of a species based solely on information about species co-occurrence (see the Methods for further details).

## Results and discussion
### From exponential to power law

Using newly available high-quality data, we reveal a surprisingly consistent relationship between co-occurrence and biotic interaction networks. Across all datasets, the proportion (*f*) of co-occurrences that correspond to actual interactions is ~20% (mean = 0.18, s.d. = 0.05; Fig. 2). The total number of potential links, as based on co-occurrence networks, ranged from 6,437 to 524, while the realized biotic interaction links ranged from 967 to 90 (Supplementary Table 1). Despite large differences in the total number of links across datasets, the proportion of potential links that are realized is strikingly similar among them. Furthermore, and also across datasets, the degree distribution in co-occurrence networks is best described by an exponential function. In contrast, realized interaction networks are best described by a power law (Fig. 2 for consumers and Extended Data Fig. 1 for resources; see also Supplementary Table 2), indicating that the proportion of realized links is not uniform across species (Fig. 3).

The prevalence of scale-free degree (that is, power law) distributions in species interaction networks is well-known in the ecological literature[17,18]. However, our results extend this understanding by demonstrating that co-occurrence networks, in contrast, exhibit exponential degree distributions. This indicates that the occurrence of a link in the network is largely independent of the presence of other links, as expected when links are randomly distributed. Yet, despite co-occurrence networks having random-like, exponential degree distributions, we reveal that assemblages of interacting species exhibit scale-free, power-law degree distributions. Such network topologies are robust to species extinctions[17,23], especially when threats are external to natural population dynamics, such as those caused by human activities[24]. Our findings thus raise the critical question: How does the exponential degree distribution in the co-occurrence network transform into a power law when we remove co-occurrence links that do not correspond to actual biotic interactions?

### The role of super-generalist species

For the degree distribution to shift from an exponential to a power law, the proportion of realized biotic interactions cannot be uniform across species. If the interaction degree of species (that is, realized biotic interactions) was roughly proportional to the co-occurrence degree (that is, potential interactions), then the respective degree distributions would keep the same functional form: an exponential would remain an exponential and a power law would remain a power law (grey line in Fig. 3; see Supplementary Text 2 for a mathematical demonstration). The interaction degree distribution would thus be a simple rescaling of the one for co-occurrences (see below where we discuss an actual null model of this kind). This is not what is seen in the empirical data. We find that the number of realized interactions increases superlinearly with the number of potential interactions (Fig. 4 for consumers and Extended Data Fig. 2 for resources). This means, in particular, that species co-occurring with more potential interacting partners keep a higher proportion of those links as biotic interactions than species with fewer co-occurrences. The relationship between species' ability to occupy multiple sites (indicative of co-occurrence with many different species) and its capacity to interact with multiple potential resources (indicative of having broad diets) has been previously recognized[26] and is critical for understanding the patterns observed in this study (Fig. 3). The fact that the most generalist species in terms of habitat use (abiotic niche) interact with a disproportionately larger number of their potential resources (biotic niche) promotes the emergence of the fat tail in the degree distribution of the biotic interaction network (Supplementary Text 2), which means that there is a higher probability of finding species in the network with a large number of links (yellow line in Fig. 3). These super-generalist species thus have a crucial role in understanding community assembly, chiefly the emergence of power-law degree distributions in the realized network of interactions.

Besides their structuring role in community assembly, the existence of super-generalist species is important for conservation. The size of species' geographical ranges is a well-known determinant of species vulnerability to extinction[27–29]. We demonstrate a general and predictable relationship between the spatial distribution of species and their biotic interactions. Widespread species, with large geographical ranges, have a vast array of interacting species, which makes them less

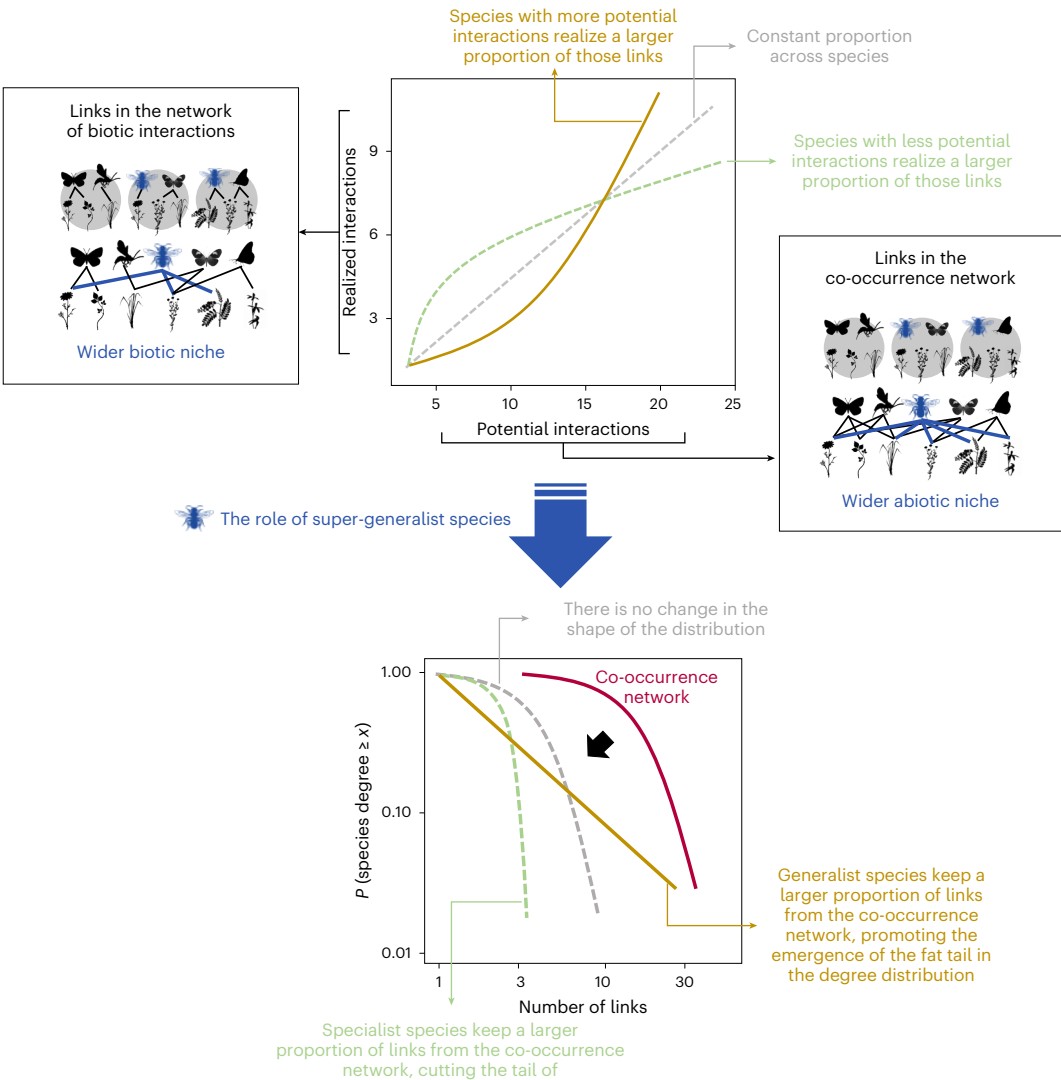

**Fig. 3 | The role of super-generalist species in the emergence of power-law degree distributions.** Top, different possibilities for the relationship between the number of potential interactions and the number of realized links in the network of biotic interactions. The yellow line shows the pattern observed in the data analysed in this study, where generalist species in terms of potential interactions realize a disproportionate large number of those links in the network of biotic interactions. The non-linearity of this relationship makes the non-proportionality across species explicit. This relationship between potential and realized (biotic) interactions highlights the strong generalism of the species both in terms of their abiotic niche (that is, larger occupancy in space and thus larger co-occurrence with other species) and their biotic niche (that is, more biotic interactions among those co-occurring with them). We call these species super-generalists. The grey and green dashed lines represent two other possible cases: a constant proportion of realized links across species and the case where specialist species would realize a larger proportion of the potential links. Bottom, the consequences of these patterns for the shift in the network degree distribution from co-occurrence to biotic interactions. Given that super-generalist species keep a larger proportion of their potential links than specialist species, the degree distribution changes from an exponential in the co-occurrence network to a power law in the network of biotic interactions (yellow line), where the probability of finding a species in the network with a large number of links is higher than in the other cases (grey and green dashed lines). If the interaction degree of species was roughly proportional to the co-occurrence degree, the respective degree distributions would keep the same functional form. The interaction degree distribution would thus be a simple rescaling of the one for co-occurrences. An exponential would remain an exponential, and a power law would remain a power law (see Supplementary Text 2 for a mathematical demonstration).

vulnerable to potential extinctions of their interacting partners. On the other hand, small-ranged species are not only threatened because of their restricted range size but also because they have a restricted pool of interacting partners.

## Frequency of co-occurrences as a key predictor

The 'super-generalist' pattern can be elucidated further. Across datasets, we observed that species co-occurring with more potential resources (or consumers if we take the resource perspective) also co-occur with them more frequently (Fig. 5 for consumers and Extended Data Fig. 3 for resources). Based on this finding, and to predict the degree distributions of interaction networks, we propose a simple interaction rate (IR) model to prune the co-occurrence network. This model uses the frequency of co-occurrences between species to predict interactions. Specifically, the IR model makes a probabilistic prediction for the interaction between species based on a uniform per-site probability of interaction ($p$). The more frequently two species co-occur, the more likely the model will assign an interaction between them (Methods). In essence, the IR model provides an expected number of interactions for each species of the

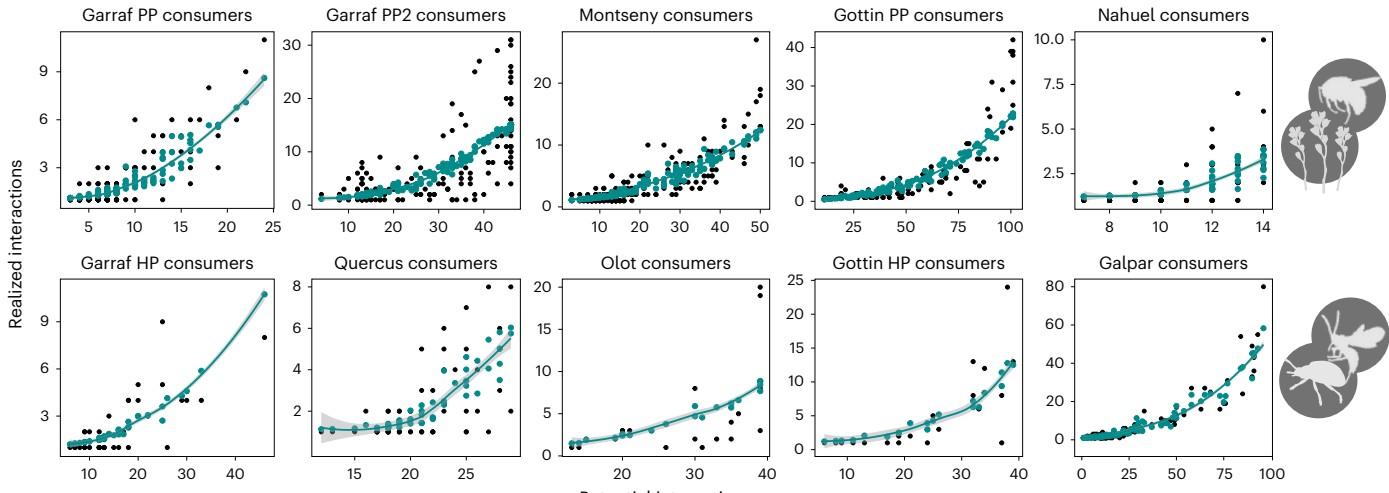

**Fig. 4 | Relationship between the number of potential interactions (based on co-occurrences) and the number of realized interactions for consumer species.** Top, PP networks. Bottom, HP networks. Each black point represents a species in the empirical system; the green line and points indicate the predicted proportion of realized interactions by the proposed model based on the frequency of co-occurrences. Notice that the relationship is non-linear, demonstrating that the proportion of co-occurrence interactions realized in biotic interactions is not constant across species. The green lines represent the mean tendency and the shaded areas represent the 95% confidence intervals. See the corresponding results for resource species in Extended Data Fig. 2. A description of all datasets can be found in Supplementary Text 1.

network, based on the frequency of their co-occurrences with potential interaction partners.

At the species level, the results of the IR model correlate relatively well with the actual realized number of interactions of each species (Extended Data Fig. 4). The IR model predicts a superlinear relationship between the number of potential and realized interactions (green line in Fig. 4 and Extended Data Fig. 2) like the one observed in the empirical patterns. On the other hand, at the community level, the IR model provides a prediction for the degree distribution of the interaction network that is strikingly accurate, regardless of whether the focus is on consumers (indegree, Fig. 6, but see below where we discuss small deviations) or resources (outdegree, Extended Data Fig. 5; in this case the prediction is near-perfect). Even though the interaction between two given species may be hard to predict based on co-occurrence alone, we found that accurate inference is possible when we shift the focus towards community level properties, such as the degree distribution of the interaction network. That being said, to predict features of the network of biotic interactions from co-occurrence frequencies, we had to estimate the per-site IR $p$. In our analysis, we estimated $p$ for each dataset by asking if the expected number of links of the pruned co-occurrence network equated the actual number of biotic interactions (Methods). While we made use of all the knowledge available to derive $p$, it is worth noting that $p$ can be seen as the average IR over species. Like all aggregate features, this means that it can be estimated from partial knowledge, such as the rate of interaction of a few randomly chosen species. Furthermore, we found that the per-site IR $p$ did not vary strongly across datasets (mean = 0.077, s.d. = 0.028) (Supplementary Table 2). This consistency allows generating predictions about the number of links and their distribution among species in biotic interaction networks using co-occurrence data, without requiring additional information.

While the IR model prediction for the degree distribution of the network of biotic interactions is accurate, there is still a small but systematic deviation when taking the consumer's perspective: it either matches or underestimates the probability associated with the species with the highest degree. That is, it underestimates the number of realized interactions of the most generalist consumers. In Fig. 6, this is most visible for Garraf PP2 and Gottin PP networks, or for the Olot HP network. This bias indicates that these super-generalist consumers have even more biotic interactions than what would be expected based on their frequency of co-occurrence with potential resources. This observation suggests the existence of a positive feedback between the biotic and abiotic niches. Initially, one might assume that species with large ranges simply have more opportunities to interact with more resources, leading to their generalist behaviour. However, the fact that the prediction still underestimates the realized biotic niche of already highly generalist species, suggests that causality might be reversed. It is possible that species capable of occupying many locations (large abiotic niche) are the ones with a natural tendency to be generalists (large biotic niche). That the model deviation only occurs when taking the consumer's perspective corroborates this interpretation.

**Beyond random pruning of co-occurrences**

To further demonstrate the importance of considering the frequency of co-occurrence to accurately estimate the degree distribution of biotic interaction networks, we conducted a null model based on a random pruning of links. In this model, the probability of interaction between a pair of species is not influenced by their frequency of co-occurrence. Instead, all co-occurring species from opposite trophic levels have the same probability of interacting, which is equal to the observed proportion ($f$) of co-occurrence links that correspond to actual interactions. Thus, the resulting network of biotic interactions is a random subset of the co-occurrence network. The degree distributions obtained from this random pruning model do not exhibit the patterns observed in the empirical data (Extended Data Fig. 6). Instead, the shape of the degree distribution of the pruned networks is similar to the one observed in the co-occurrence networks, as suggested in Fig. 3, and the simple mathematical argument that explains why the two degree distributions will only be rescaled versions of each other (Supplementary Text 2). This difference between the results obtained from the IR model and the random pruning model highlights the crucial role of super-generalist species in the emergence of power-law degree distributions. The presence of super-generalist species that realize a disproportionately large number of their potential links leads to the power-law degree distributions observed in nature. In contrast, the absence of such super-generalist species in the random pruning model results in degree distributions that do not resemble the ones observed in real ecological communities.

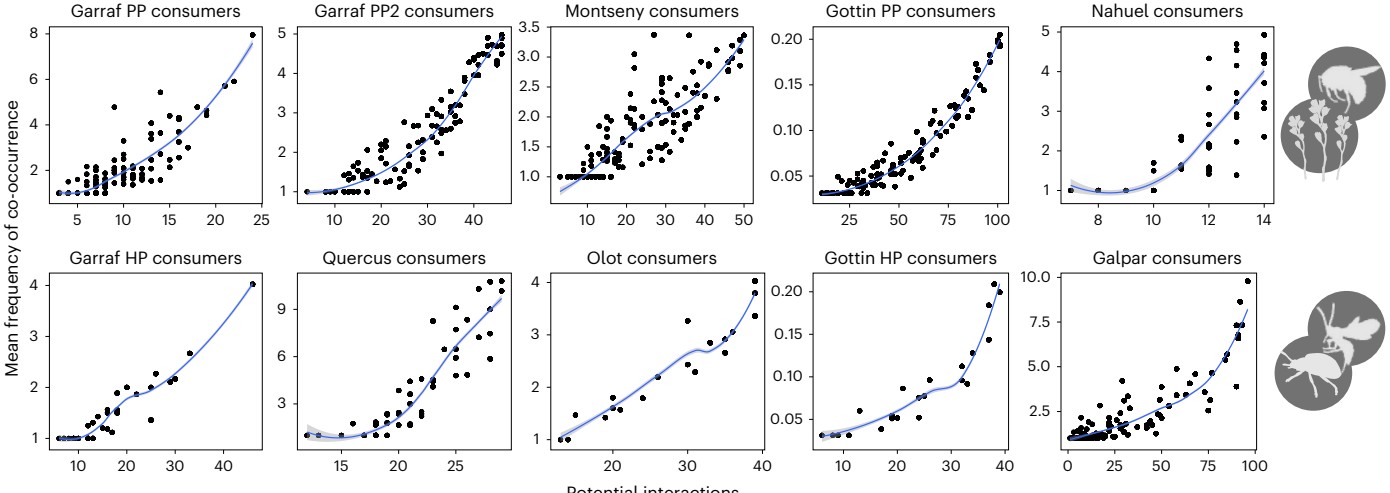

**Fig. 5 | Relationship between the number of potential interactions of consumer species and the average frequency of co-occurrence with their potential resources.** Top, PP networks. Bottom, HP networks. Each black point represents a species in the empirical system. The blue lines represent a gam fit only for visualization purposes; the shaded areas represent the 95% CIs. A description of all datasets can be found in Supplementary Text 1. See the corresponding results for resource species in Extended Data Fig. 3.

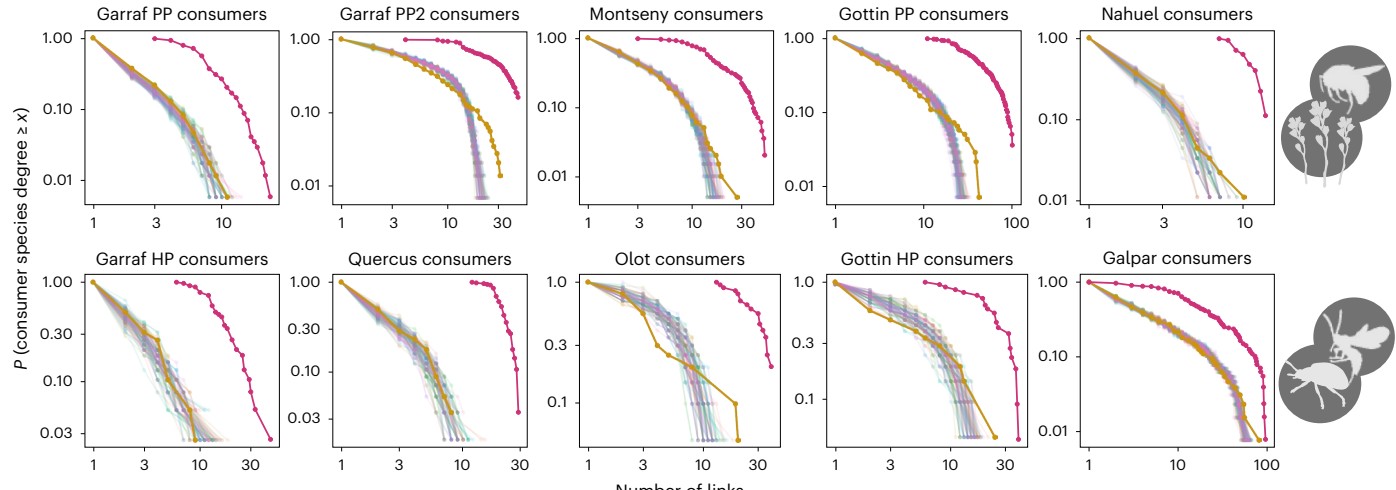

**Fig. 6 | Model predictions for the degree distribution of consumer interactions based on the frequency of species co-occurrences.** The model assumes a fixed probability of interaction for each site where a co-occurrence between consumer and resource is observed. Thus, given that it is a probabilistic model, different random realizations are depicted in shaded colours. The actual degree distribution of the network of biotic interactions is shown in yellow and the degree distribution of the co-occurrence network is shown in red. Top, PP networks. Bottom, HP interactions. See the corresponding results for resource species in Extended Data Fig. 5.

## Conclusion

Characterizing the intricate web of interactions among multiple organisms co-occurring in ecosystems is a challenging task for current science. A much desired approach is inferring biotic interactions from species co-occurrence data[1,3,4]. While accurately detecting pairwise interactions from co-occurrence data is difficult[13,14], we discovered that analysing co-occurrence patterns at the network level can yield valuable information for inferring the biotic interaction structure of ecological communities.

Our research unveiled a clear relationship between the two types of networks that is remarkably consistent across datasets. Notably, we found that super-generalist species[26], possessing broad environmental tolerance and diet generality, have a key role in structuring ecological communities. These super-generalists interact with a disproportionately large number of co-occurring species, leading to a shift in the degree distribution from exponential in the co-occurrence network to a scale-free power law in the network of realized interactions. This interplay between the spatial distribution of species and their biotic interactions holds important implications for designing conservation strategies at large spatial scales.

To move beyond observational comparisons between co-occurrence and interaction networks, we demonstrate how accounting for the frequency of co-occurrence between species enables accurate prediction of fundamental properties of the realized network of interactions, explaining the observed patterns effectively. In essence, we found that species not only need to co-occur to interact, but need to co-occur frequently. Our findings represent an important advancement in understanding community assembly while providing unexpectedly

realistic predictions of biotic interaction networks based on species co-occurrence information alone. However, it is important to acknowledge the influence of spatial scale when predicting biotic interaction networks from co-occurrences. The spatial scale at which species co-occurrences are assessed can impact the comparison between these two types of networks. Generally, coarser scales may require stronger pruning of the co-occurrence network to approximate the interaction network. In this study, we used the same spatial scale to assess co-occurrences and characterize biotic interactions, allowing for direct comparisons. Nonetheless, future research should delve into how this comparison changes with increasing spatial scale and whether there is a threshold beyond which co-occurrences no longer serve as accurate predictors of biotic interaction networks.

Given the challenges in sampling biotic interactions across large spatial extents, many studies resort to different approaches to infer species interactions, such as expert knowledge, literature reviews or proxies like species traits[30–34]. The generalities revealed in this study can inform and enhance these approaches to better predict interaction network features based on co-occurrence data, substantially simplifying the effort required. Specifically, the per-site IR ($p$) found in this study can serve as a valuable reference for generating predictions on biotic interaction networks without the need for additional information on biotic interactions. Moreover, $p$ is strongly correlated with the proportion of co-occurrence interactions that are actual biotic interactions ($f$) (Extended Data Fig. 7). Because estimating the per-site IR is easier in empirical data than determining the full network of biotic interactions, this correlation can provide valuable insights. Yet, further research is needed to extend the framework presented in this study beyond bipartite networks (for example, food webs or competitive networks), and to not only predict the basic properties of realized interaction networks from co-occurrence but to predict the interactions themselves. Advancements in this direction would substantially improve our understanding of the complex web of interactions within ecological communities and facilitate more informed conservation and management strategies. The findings of this study open up exciting opportunities to explore and refine the use of co-occurrence data for predicting species interactions and advancing our knowledge of community dynamics at several spatial scales.

## Methods

### Empirical data
We used ten empirical datasets of bipartite networks comprising both mutualistic and antagonistic interactions. All datasets contained information about species distributions across multiple sampled locations and their interactions in each location. Specifically, we used datasets describing PP interactions from forests in the natural park of Montseny[35], from Mediterranean shrublands in Garraf (two datasets[36,37]), from a temperate forest in Argentinian Patagonia[38] and from calcareous grasslands in central Germany[39]. We used datasets describing HP interactions from the natural park of Olot in Catalonia[40], from Mediterranean shrublands in Garraf[36,37], from a temperate forest in Finland[41], from calcareous grasslands in central Germany and from a dataset spanning a large latitudinal gradient from Italy to northern Norway[42]. All datasets are explained in more detail in Supplementary Text 1 and are available in the article by Galiana et al.[43]. The description of the basic network properties of each dataset can be found in Supplementary Table 1.

### Network construction
For each dataset, we built the network of co-occurrences and the network of biotic interactions based on empirical observations (Fig. 1a). The co-occurrence network was built from the empirical observations of the species' spatial distribution in each dataset. Therefore, a link between two species from different trophic levels was added when they co-occurred in at least one spatial unit of each dataset. In addition to the binary co-occurrence information (that is, presence or

absence of co-occurrence between species) used to build the network of co-occurrences, we also considered how frequently species from different trophic levels co-occurred across sites to develop our theoretical model (see section below). The network of species co-occurrences has been interpreted as a network of potential interactions given that to interact species normally have to coexist, with the possible exception of strongly negative interactions leading to competitive co-exclusion[2]. Given that the datasets considered in this study describe PP and HP interactions, species indeed need to co-occur to interact.

The network of biotic interactions is based on empirical observations of the ecological interactions between species in each spatial unit. Thus, an interaction between two species was added to the network of biotic interactions if it was empirically observed in at least one spatial unit. It is important to note that the spatial scale considered for the biotic interactions is the same as the one considered for co-occurrences. That is, the presence or absence of interactions and co-occurrences between species was empirically observed in each spatial unit of each dataset. The dimensions of the spatial units of each dataset were chosen by the authors of the original papers describing these datasets[35–43] to ensure the correct description of the network of biotic interactions. Therefore, if two species were observed together but not interacting, we can be confident that it is not a false negative. The use of the same spatial scale for co-occurrences and interactions allowed us to directly compare the two.

### Network degree distribution
Both the co-occurrence and biotic interaction networks were described through quantification of their degree distribution in the empirical data. The degree distribution is defined as the probability $P(k > x)$ of (uniformly) choosing a species that has at least $x$ links to other species in the network (Fig. 1b and Supplementary Text 2). We then fitted four different functions previously identified in ecological network degree distributions: exponential; power law; truncated power law; and log-normal[17,44]. After fitting these functions to the degree distribution of both co-occurrence and biotic interaction networks for each dataset, we selected the most parsimonious function using the Akaike information criterion. Because the networks analysed are bipartite, we could analyse consumer and resource degree distributions independently; we present the results for each trophic level separately.

### Pruning of the co-occurrence network based on the frequency of co-occurrence
To investigate the differences between the degree distributions of the two network types (that is, co-occurrence and biotic interactions) and examine how co-occurrence networks can be pruned to approximate the realized network of biotic interactions, we propose a simple model based on the frequency of co-occurrence between species (that is, the number of times two species co-occur across sampling units) to estimate the expected number of interactions of a species.

Let $N = (N_{\alpha i})$ be the frequency of the co-occurrence matrix where $N_{\alpha i}$ is the number of sites where a consumer $\alpha$ has been observed together with a resource $i$. Given a rate of interaction per site $p$, the probability that $\alpha$ and $i$ actually interact is:

$$P_{\alpha i} = 1 - (1-p)^{N_{\alpha i}}$$

Therefore, $P_{\alpha i}$ is the probability of the interaction between consumer $\alpha$ and resource $i$, which is modulated by the frequency of co-occurrence between both species across sites $N_{\alpha i}$.

We still need to fix the value for $p$. To do so, we may note that over many random draws of the model, the expected number of links, $L$, conditioned on the fact that all consumers have at least one resource is:

$$E(L(p)) = \sum_{\alpha} \sum_{i} \frac{1 - (1-p)^{N_{\alpha i}}}{1 - (1-p)^{N_{\alpha}}}$$

where $N_\alpha = \sum_i N_{\alpha i}$. We imposed such a condition to ensure that we did not have isolated consumers lacking a required resource to survive (as observed in the empirical data). We then defined $p$ so that $\frac{E(L(p))}{L_c} = f$, which imposes that, on average over random realizations of the model, the total number of links $L$ coincides with the number of actual links in the network of biotic interactions, which is a proportion $f$ of the total number of co-occurrences, that is, $L_c$ ($f$ corresponds to the percentages shown in Fig. 2). Thus, for each dataset, $p$ is adjusted to generate a pruned network that has, on average, the same number of biotic interactions than the original network. In the end, any random realization of this model generates the number of links for each species, which allowed us to characterize the degree distribution and compare it to the actual degree distribution of the empirical network of biotic interactions. Given that it is a probabilistic model, we performed 100 different random realizations of the model (the many coloured lines in Fig. 5).

### Random pruning of the co-occurrence network

To demonstrate the importance of considering the frequency of co-occurrences between species (which underlie the IR model proposed earlier) to properly infer network degree distributions, we additionally performed a random pruning of the network of co-occurrences and compared the results. That is, while in the model proposed earlier the probability of interaction between a pair of species depends on their frequency of co-occurrence in space, in the random pruning model all species from opposite trophic levels that co-occur have the same probability of interacting. For instance, if for a given dataset we observe that 20% of the links from the co-occurrence network are transformed into biotic interactions, all species that co-occur will have a probability of interacting $P_{\alpha i} = 0.2$, which is equivalent to set $N_{\alpha i} = 1$ for all species in the model described earlier. We performed 100 replicates of this random pruning and compared the resulting degree distributions with the empirically observed degree distributions of the biotic interaction networks.

### Reporting summary

Further information on research design is available in the Nature Portfolio Reporting Summary linked to this article.

## Data availability

The data supporting the study results can be found at https://doi.org/10.5281/zenodo.8402455.

## Code availability

The code used to analyse the data and generate the results can be found at https://doi.org/10.5281/zenodo.8402455.

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

## Acknowledgements

N.G. received funding from the European Union's Horizon 2020 research and innovation programme under the Marie Skłodowska-Curie grant agreement BIOFOODWEB (no. 101025471). M.B.A. acknowledges funding from the Spanish Ministry of Science, Innovation and Universities through the PredWeb project (PGC2018–099363-B-I00) and, together with A.R., from the European Union's Horizon 2020 research and innovation programme under grant agreement AQUACOSM-Plus (no. 871081). J.-F.A. was supported by the Laboratoires d'Excellences (LABEX) TULIP (ANR-10-LABX-41).

## Author contributions
N.G., J.-F.A. and M.B.A. designed the research. N.G. performed the research and analysed the data. J-F.A. and N.G. developed the theoretical framework. N.G. wrote the manuscript with feedback from J.-F.A. and M.B.A., and comments from F.M. and A.R.

## Competing interests
The authors declare no competing interests.

## Additional information
**Extended data** is available for this paper at https://doi.org/10.1038/s41559-023-02254-y.

**Correspondence and requests for materials** should be addressed to Nuria Galiana.

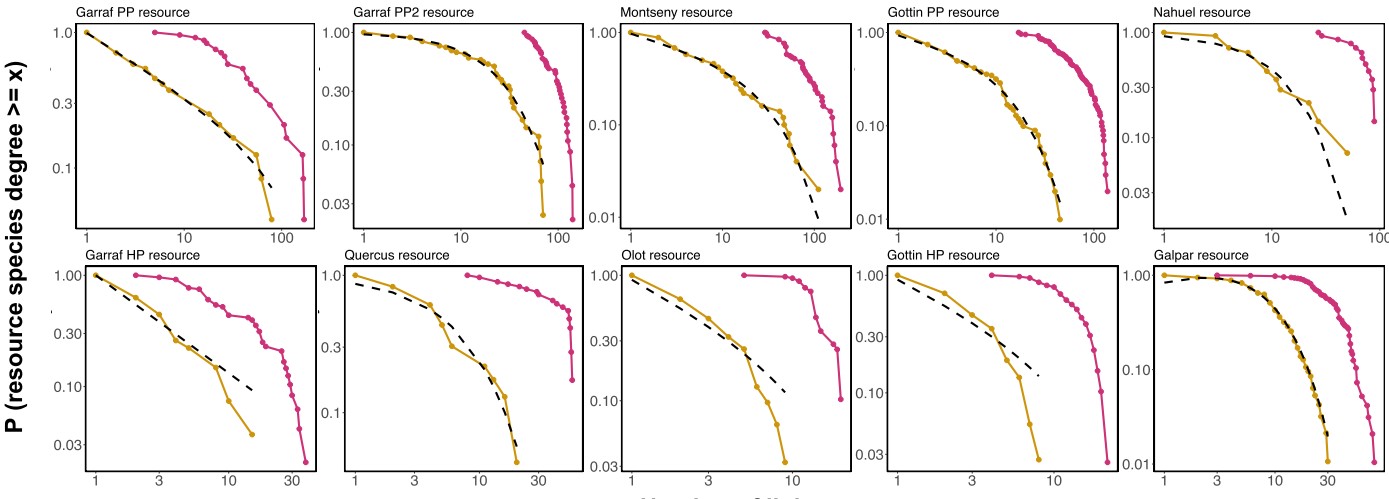

**Extended Data Fig. 1 | Comparison of the frequency distributions of resources degree among co-occurrence networks and realised biotic interactions networks.** Red lines represent co-occurrence networks and yellow lines represent realised biotic interactions networks in the 10 datasets investigated. Top row are plant-pollinator networks and bottom row are host-parasite networks. Black dashed lines indicate the more parsimonious functions among all tested (see Methods).

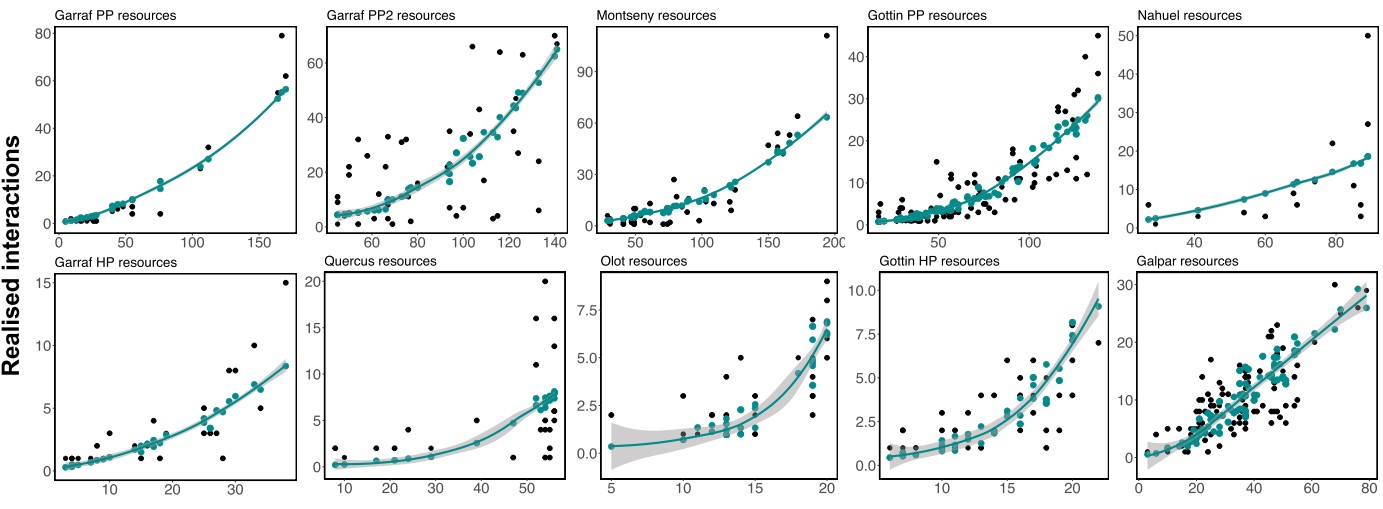

**Extended Data Fig. 2 | Relationship between the number of potential interactions and the number of realised interactions and model prediction for resource species.** Top row are plant-pollinator networks and bottom row are host-parasite interactions. Each black point represents a species in the empirical system and the green line and points indicate the predicted proportion of realised interactions by the proposed model based on the frequency of co-occurrences. Green lines represent the mean tendency and shaded areas represent 95% confidence intervals.

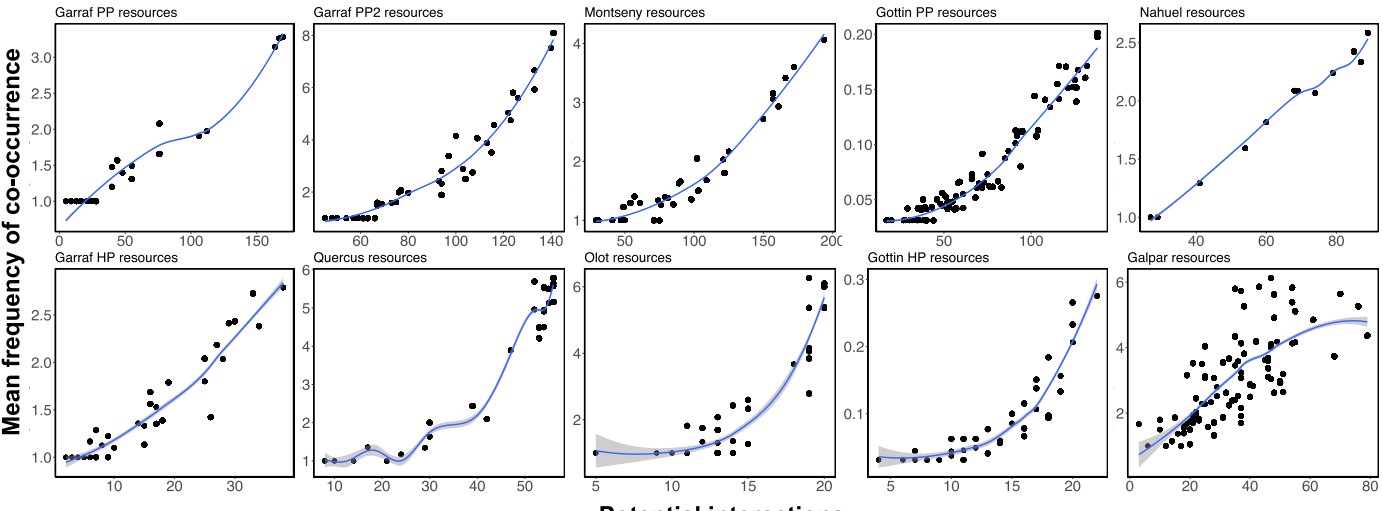

**Extended Data Fig. 3 | Relationship between number of potential interactions for resource species and the mean frequency of co-occurrence with their consumers.** Top row are plant-pollinator networks and bottom row are host-parasite interactions. Each black point represents a species in the empirical system. Blue lines represent a *gam* fit only for visualisation purposes and shaded areas represent 95% confidence intervals.

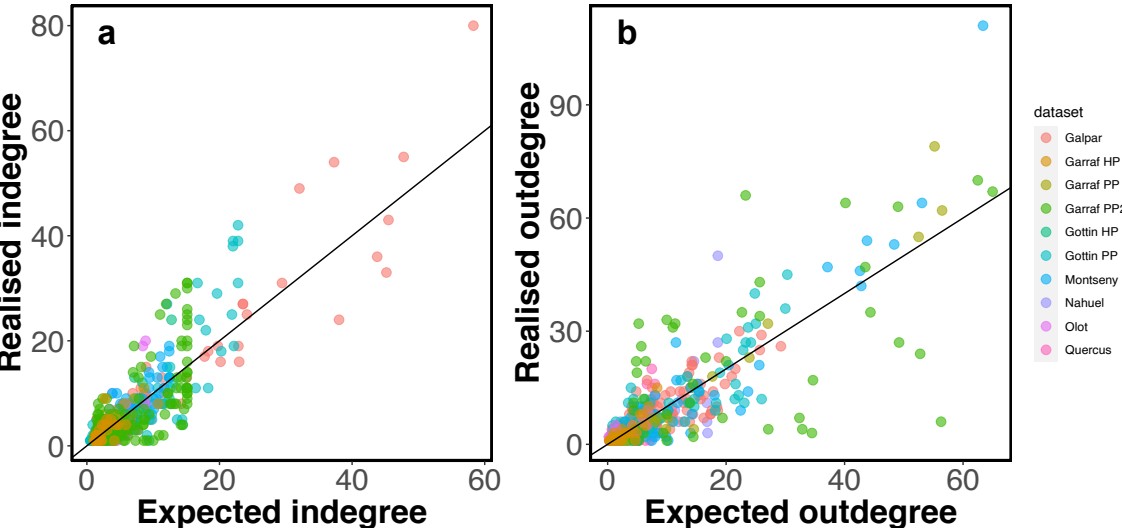

**Extended Data Fig. 4 | Relationship between the number of expected interactions based on the frequencies of species co-occurrences and the number of realised interactions for each species.** (**a**) Shows the relationship from the consumer's perspective (that is indegree is the number of resources each consumer has) while (**b**) represents the resources perspective (that is outdegree corresponds to the number of consumers each resource has). Each point represents a species in the empirical system. Black line shows the 1:1 line indicating a perfect relationship between the predicted and the realised number of interactions.

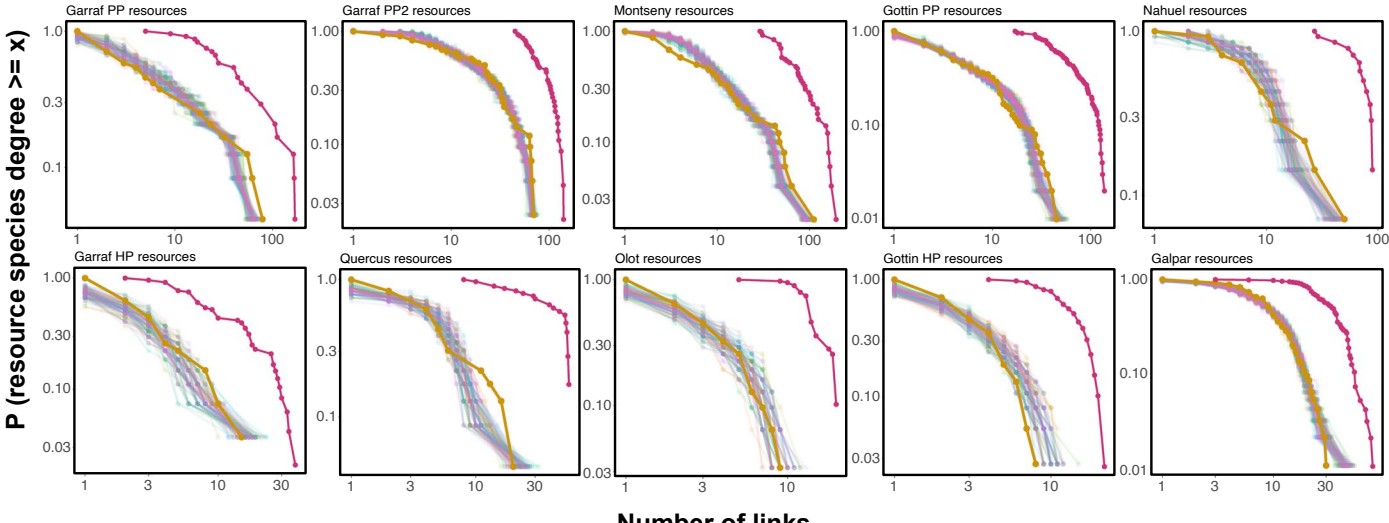

**Extended Data Fig. 5 | Comparison of the degree distributions of resource species among co-occurrence networks, realised biotic interactions networks and our theoretical predictions accounting for the frequency of interactions.** Red lines represent co-occurrence networks, yellow lines represent realised biotic interactions networks and our theoretical predictions accounting for the frequency of interactions are represented in multicolor (each color represents a replicate) in the 10 datasets investigated. Top row are plant-pollinator networks and bottom row are host-parasite networks.

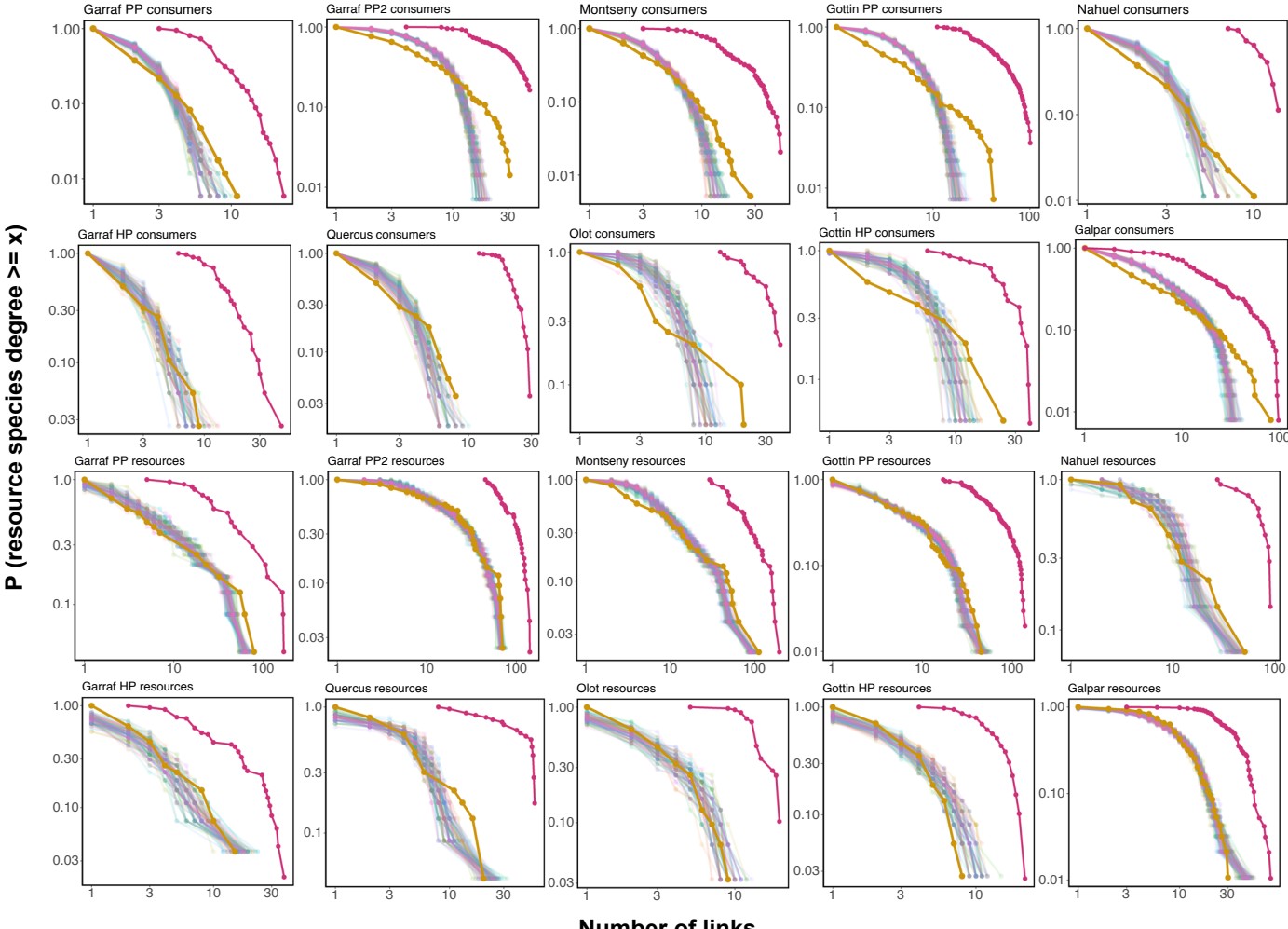

**Extended Data Fig. 6 | Comparison of the degree distributions among co-occurrence networks, realised biotic interactions networks and the null model predictions.** Red lines represent co-occurrence networks, yellow lines represent realised biotic interactions networks and the null model predictions are represented in multicolor (each color represents a replicate) for the 10 datasets investigated. The null model prunes the co-occurrence networks using a constant proportion of links to keep across species. Therefore, it results in a random pruning of the co-occurrence network. Top rows correspond to consumers and bottom rows correspond to resource species.

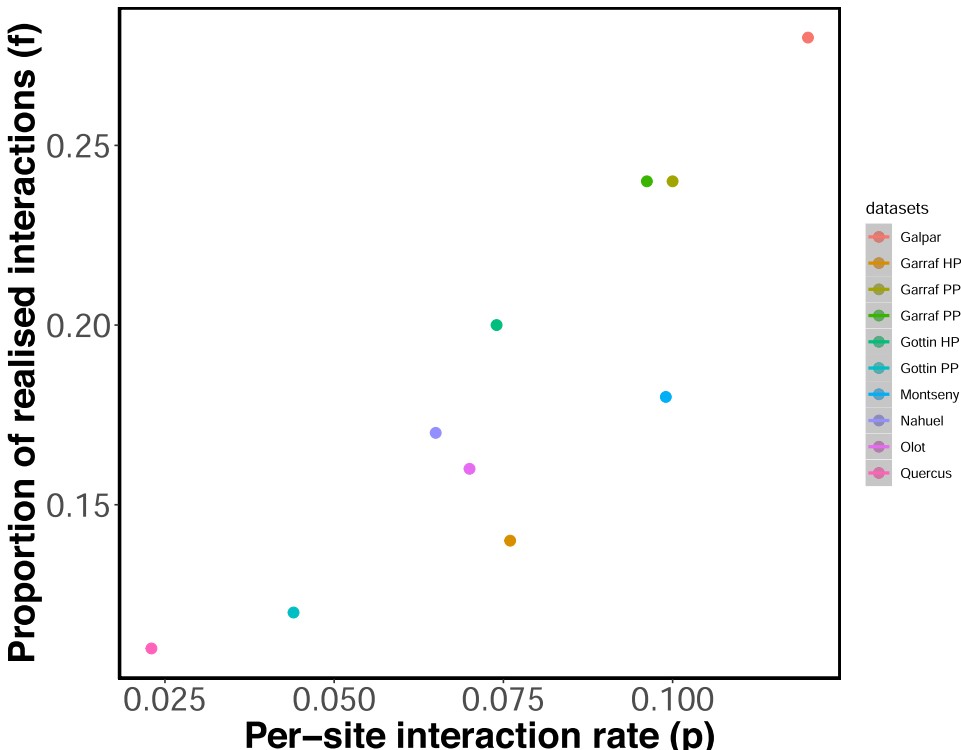

**Extended Data Fig. 7 | Relationship between the per-site interactions rate (p) and the proportion of realised links (f) across datasets.** Each color represents a different dataset.

# Reporting Summary

## Statistics

For all statistical analyses, confirm that the following items are present in the figure legend, table legend, main text, or Methods section.

| n/a | Confirmed | |
|---|---|---|
| ☐ | ☒ | The exact sample size ($n$) for each experimental group/condition, given as a discrete number and unit of measurement |
| ☐ | ☒ | A statement on whether measurements were taken from distinct samples or whether the same sample was measured repeatedly |
| ☐ | ☒ | The statistical test(s) used AND whether they are one- or two-sided <br> *Only common tests should be described solely by name; describe more complex techniques in the Methods section.* |
| ☐ | ☒ | A description of all covariates tested |
| ☐ | ☒ | A description of any assumptions or corrections, such as tests of normality and adjustment for multiple comparisons |
| ☐ | ☒ | A full description of the statistical parameters including central tendency (e.g. means) or other basic estimates (e.g. regression coefficient) AND variation (e.g. standard deviation) or associated estimates of uncertainty (e.g. confidence intervals) |
| ☐ | ☒ | For null hypothesis testing, the test statistic (e.g. $F$, $t$, $r$) with confidence intervals, effect sizes, degrees of freedom and $P$ value noted <br> *Give P values as exact values whenever suitable.* |
| ☒ | ☐ | For Bayesian analysis, information on the choice of priors and Markov chain Monte Carlo settings |
| ☒ | ☐ | For hierarchical and complex designs, identification of the appropriate level for tests and full reporting of outcomes |
| ☐ | ☒ | Estimates of effect sizes (e.g. Cohen's $d$, Pearson's $r$), indicating how they were calculated |

*Our web collection on statistics for biologists contains articles on many of the points above.*

## Software and code

Policy information about availability of computer code

| Data collection | No software was used for data collection. |
|---|---|
| Data analysis | The code used to analyse the data and generate the results can be found in the following repository: DOI: 10.5281/zenodo.8402455 |

For manuscripts utilizing custom algorithms or software that are central to the research but not yet described in published literature, software must be made available to editors and reviewers. We strongly encourage code deposition in a community repository (e.g. GitHub). See the Nature Portfolio guidelines for submitting code & software for further information.

## Data

Policy information about availability of data

All manuscripts must include a data availability statement. This statement should provide the following information, where applicable:
- Accession codes, unique identifiers, or web links for publicly available datasets
- A description of any restrictions on data availability
- For clinical datasets or third party data, please ensure that the statement adheres to our policy

The data supporting the results can be found in: DOI: 10.5281/zenodo.8402455

# Research involving human participants, their data, or biological material

Policy information about studies with human participants or human data. See also policy information about sex, gender (identity/presentation), and sexual orientation and race, ethnicity and racism.

| | |
|---|---|
| Reporting on sex and gender | N/A |
| Reporting on race, ethnicity, or other socially relevant groupings | N/A |
| Population characteristics | N/A |
| Recruitment | N/A |
| Ethics oversight | N/A |

Note that full information on the approval of the study protocol must also be provided in the manuscript.

# Field-specific reporting

Please select the one below that is the best fit for your research. If you are not sure, read the appropriate sections before making your selection.

☐ Life sciences ☐ Behavioural & social sciences ☒ Ecological, evolutionary & environmental sciences

For a reference copy of the document with all sections, see nature.com/documents/nr-reporting-summary-flat.pdf

# Ecological, evolutionary & environmental sciences study design

All studies must disclose on these points even when the disclosure is negative.

| | |
|---|---|
| Study description | We use 10 well-resolved empirical bipartite networks from terrestrial habitats across the globe that contain information about both species distributions across multiple locations and their empirically-sampled biotic interactions (both mutualistic and antagonistic). Such newly available datasets allow us to directly compare, for the first time, co-occurrence and interaction networks, and we do so in terms of their degree distribution. We then examine how co-occurrence networks should be pruned to approximate features of the realised network of biotic interactions. We propose a simple model that uses knowledge not only on co-occurrence between species, but also on the frequency of such co-occurrence (that is, the number of sites where co-occurrence is known) to estimate the number of interactions of a species, and thus to predict the degree distribution of interaction networks based on information of species co-occurrence alone. |
| Research sample | The study uses 10 existing datasets involving different types of species interactions from different ecosystems and biomes, and spanning different spatial scales. All data used is already publicly available. |
| Sampling strategy | Each of the datasets used, followed different sampling strategies depending on the system analysed. Details about the sampling strategy of each dataset used are briefly explained in the methods section of this manuscript and appendix and explained in depth in the Nature Ecology and Evolution paper where the data was originally published (as explained in the text). |
| Data collection | The data collection was also specific of each dataset used. |
| Timing and spatial scale | The spatial scale differs across datasets but we take this into account given that we compare the network of co-occurrence with the network of interactions for each dataset. |
| Data exclusions | No data was excluded from the analyses. |
| Reproducibility | The study is not based on experimental findings. It is based on species interactions networks of empirical communities sampled in natural systems. |
| Randomization | We performed randomization to analyse the output of the proposed theoretical to prune the co-occurrence network. |
| Blinding | Blinding is not relevant in our study given that the data used describes species interaction networks based on empirical observations of natural systems and are not based on experimental communities in the lab. |

Did the study involve field work? ☐ Yes ☒ No

# Reporting for specific materials, systems and methods

We require information from authors about some types of materials, experimental systems and methods used in many studies. Here, indicate whether each material, system or method listed is relevant to your study. If you are not sure if a list item applies to your research, read the appropriate section before selecting a response.

## Materials & experimental systems

| n/a | Involved in the study |
|---|---|
| ☒ ☐ | Antibodies |
| ☒ ☐ | Eukaryotic cell lines |
| ☒ ☐ | Palaeontology and archaeology |
| ☒ ☐ | Animals and other organisms |
| ☒ ☐ | Clinical data |
| ☒ ☐ | Dual use research of concern |
| ☒ ☐ | Plants |

## Methods

| n/a | Involved in the study |
|---|---|
| ☒ ☐ | ChIP-seq |
| ☒ ☐ | Flow cytometry |
| ☒ ☐ | MRI-based neuroimaging |

