## [Peer Review File · Nature Ecology & Evolution]

Peer Review Information

Journal: Nature Ecology & Evolution

Manuscript Title: Power-laws in species' biotic interaction networks can be inferred from co-occurrence data

Corresponding author name(s): Nuria Galiana

Editorial Notes:

Reviewer Comments & Decisions:

Decision Letter, initial version:

15th June 2023

Dear Dr Galiana,

Your Article, "From species co-occurrences to interactions: the emergence of power-law degree distributions" has now been seen by three reviewers. You will see from their comments copied below that while they find your work of considerable potential interest, they have raised quite substantial concerns that must be addressed. In light of these comments, we cannot accept the manuscript for publication, but would be very interested in considering a revised version that addresses these serious concerns.

We hope you will find the reviewers' comments useful as you decide how to proceed. If you wish to submit a substantially revised manuscript, please bear in mind that we will be reluctant to approach the reviewers again in the absence of major revisions.

There is consensus among the reviewers that while the analyses are well done, but more clarity is needed in presenting the methods, and that the overarching hypothesis and premise are not yet convincing--a major restructure will be necessary to address that.

If you choose to revise your manuscript taking into account all reviewer and editor comments, please highlight all changes in the manuscript text file [OPTIONAL: in Microsoft Word format].

* Include a "Response to reviewers" document detailing, point-by-point, how you addressed each referee comment. If no action was taken to address a point, you must provide a compelling argument. This response will be sent back to the referees along with the revised manuscript.

* If you have not done so already we suggest that you begin to revise your manuscript so that it conforms to our Article format instructions at <http://www.nature.com/natecolevol/info/final->

2submission. Refer also to any guidelines provided in this letter.

[REDACTED]

If you wish to submit a suitably revised manuscript we would hope to receive it within 6 months. If you cannot send it within this time, please let us know. We will be happy to consider your revision so long as nothing similar has been accepted for publication at Nature Ecology & Evolution or published elsewhere.

Nature Ecology & Evolution is committed to improving transparency in authorship. As part of our efforts in this direction, we are now requesting that all authors identified as 'corresponding author' on published papers create and link their Open Researcher and Contributor Identifier (ORCID) with their account on the Manuscript Tracking System (MTS), prior to acceptance. This applies to primary research papers only. ORCID helps the scientific community achieve unambiguous attribution of all scholarly contributions. You can create and link your ORCID from the home page of the MTS by clicking on 'Modify my Springer Nature account'. For more information please visit please visit www.springernature.com/orcid.

Thank you for the opportunity to review your work.

[REDACTED]

Reviewer expertise:

Reviewer #1: spatial scaling

Reviewer #2: species co-occurrence networks

Reviewer #3: species co-occurrence networks

Reviewers' comments:

2Reviewer #1 (Remarks to the Author):

There is much to like about this paper, it is interesting, well written, and the analyses seem fascinating, as far as they go. I really want to believe it. Unfortunately, I must say that in its current form, I just don't yet believe the main premise of the study that co-occurrence is a necessary condition for interaction. In principle, I see the point. Especially things like mutualisms. However, for predator-prey, I'm less convinced. To some degree it depends on the quality of the data for the networks. And this is where I have some doubts. The current paper refers primarily to a previous NEE paper in order to find the 'details' of the networks studied. I was actually a reviewer of that previous paper as well, both for Nature and NEE. I liked it, but in that case, I also found it hard to really understand in depth the details of the datasets and food webs. Of course, it is hard to describe completely with so many datasets and so little space, but in this case, there is some important information being lost that I would need to 'believe' the proof of concept.

Here, my specific objection refers to the statement that "co-occurrence is a necessary condition for interaction" and thus that "interaction networks are a subset of co-occurrence networks". I like the idea, but in practice, I'm having trouble believing this, depending on the quality and resolution of the data (which I believe in some cases, are very coarse). The foremost example in my mind is behavioral avoidance by prey and/or local extinction caused by predators. In freshwater systems, for example, whole suites of species, including insects, amphibians and small fish never co-occur in the same pond/lake with larger fish. They are almost never found together. But this is the result of a strong interaction and a generalist predator, together with habitat heterogeneity in the distribution of predators. Same with large mammal herbivores, seed predators, and many others. Sometimes species are never found together because the interaction is too strong and the distributions are otherwise heterogeneous. These might be extreme, but I'm certain there are many other similar examples in the kinds of data used here. Anytime there are very strong interactions on some, but not all, species in the community; habitat heterogeneity in the distribution of the predator; generalist predators; I would think similar results could emerge. Even in the mutualism data, can we be certain that the link is not possible, or just not observed because the pollinator has preferences in particular locations (nesting sites, other needs) but would certainly interact with a species if circumstances allowed. I guess in both cases there is a difference between possible interactions and realized interactions, but I'm not sure these can be distinguished with the sorts of data that are compiled in these networks. There's almost no information given about how the co-occurrence networks were estimated; at what scale; with what resolution; what effort? All of these things critically influence a species co-occurrence (and likelihood of interaction). Similarly, how were interaction networks estimated?

In short, Despite my serious questions and concerns about the premise and empirical estimates of the networks, there is much to like about this paper. It is well written, and given the data, the analyses seem well done. I found the explanations of the network metrics (degree distributions) and the meaning of these relationships between the two kinds of networks compelling. That said, for the following reasons, I just don't yet believe it.

-We have tried for decades to get process from pattern; it's almost never possible. At the least, I would need to see more simulation models indicating it's possibility, and more exploration of null expectations and scenarios

-The datasets used are not well described, but it's not at all clear how the statement "interaction

3

networks are a subset of co-occurrence networks” can be true in many of these datasets. I would need much more proof before I would believe the results. It’s just not clear

Reviewer #2 (Remarks to the Author):

Comments to the Author

This is an interesting study in which authors present and discuss whether interactions between species can be inferred from co-occurrence data. Specifically, they compare networks of co-occurrence with networks of interactions. First, they found that 20% of co-occurrences correspond to interactions. Secondly, they showed that the degree distribution shifts from exponential in co-occurrence networks to scale-free in interaction networks. In general terms, I think the study is well written, and the main problem is clearly presented. My main concern is about the presentation of the mechanisms involved in the results.

1. The hypothesis of the study should be more explicitly developed. Authors said that the pruning of the co-occurrence network makes the degree distribution change from exponential to a power-law. The co-occurrence patterns could be the result of intraspecific environmental tolerance and dispersal abilities, while the interaction network could be explained by the interplay between the biotic and abiotic niches of a species. In this context, I can’t see the mechanistic understanding of the shift in the degree distribution between co-occurrence networks and interaction networks. Maybe a conceptual figure that explains how the mechanisms lead the change could be great. Additionally, how these mechanisms are incorporated in the models.

2. What about the sign of co-occurrence and interaction networks, and the mechanisms involved in the shift of the degree distribution.

3. I did not understand why the ‘super-generalists’ pattern could explain the change from an exponential to a power-law degree distribution. Please, incorporate that in the conceptual figure, or a new one showing the role of the ‘super-generalists’ species, and how or why the frequency of co-occurrences are key predictors of degree distributions.

4. In the methods author said that: “...the expected number of links, L , conditioned on the fact that all consumers have at least one resource”. This means that the richness or the size of the co-occurrence network and the interaction network should be the same. What are the implications of that?

Reviewer #3 (Remarks to the Author):

Note that I added I also provide a pdf version of my review.

4General comments

In this manuscript, the authors analyze 10 bipartite networks to scrutinize the relationship between the degree distribution of species in co-occurrence networks (also viewed here as networks of potential interactions) and the degree distribution of the corresponding networks of biotic interactions. Their results demonstrate that while the degree distribution in interaction networks is scale-free, the degree distribution in co-occurrence networks is exponential. Based on this, they propose a simple pruning model 'to approximate the realized network of biotic interactions' based on the co-occurrence network. The simple model they apply explains remarkably well how the exponential distribution of species degree observed in co-occurrence networks mapped into the scale-free degree distribution obtained in networks of biotic interactions. Based on this, they highlight the role of super-generalist species in structuring community.

The work achieved here is based on recent methodological developments and analyze high quality datasets to propose new insights on how co-occurrence networks structure inform the structure of biotic interaction networks. While I value the authors efforts, I believe major clarifications are required to better justify the importance of their findings. In what follows I summarize my main concerns around three major comments before providing further technical comments.

Overall lack of clarity in methods

I am starting with this because the two following comments may partially be explained by this lack of clarity (hopefully, not just a lack of understanding on my end). In my opinion, the methods section should be clarified given that with the level of details provided so far, I seriously doubt that one can reproduce the analysis.

First, in 'Network construction' section:

> Therefore, a link between two species from different trophic levels is added when they co-occur in a given spatial unit of each dataset.

Does the spatial unit vary from one dataset to another? If so, does it matter (see the next comment)? Are the co-occurrence networks built here binary, meaning that a co-occurrence between two given species is set to '1' as soon as there is one co-occurrence event in at least one of the sites? If so, does that mean that there is no attempt to determine whether the co-occurrence is significant?

Also,

> Spatial co-occurrence is indeed used as a first step for inference of biotic interactions from auxiliary data.

I guess it is a general statement, not something you have done here, so this might actually bring some confusion. That said, it remains not clear how the biotic interactions were built here, I think the

5information is available for all the networks because they were built that way, is that correct? Please clarify this.

Second, in 'Network degree distribution' section:

> quantification of their degree distribution, defined as the probability $P(k)$

My understanding is that you have used the degree distribution of observed networks to obtain this distribution, so here one degree distribution is made of the collection of observed degrees in one network. Am I correct? Or are there replicates that I have missed (I would say no based on the data)?

Third, in 'Pruning of the co-occurrence network' section: a 'frequency of co-occurrence' is mentioned here, so the networks are no longer binary here? I am not sure I follow here.

Last, It is quite hard to follow what models are used in figures 2-5 (and the corresponding supplementary figures) and some model details are missing.

> [...] we selected the most parsimonious function using the Akaike information criterion (AIC)

This pertains to the models used to generate the dashed line in Figure 2, right? What about the GAM used in Figure 4 (and S3), and why using GAM there instead of simpler models?

In figures 3 and 5, my understanding is that you are using your pruning model, correct? Two question, how are the `p` for those models? In the code I have seen values of p

```
``R
# Garaf
p <- 0.07

# Nahuel
p <- 0.065

# Gottin
p <- 0.074

# Quercus
p <- 0.0227

# Salix
p <- 0.121
``
```

but I am not quite sure how they are obtained. Also, in figure 5, I do not understand how the different lines were obtained. It is mentioned that they are random realizations, but it is not clear to me what does it mean in this context. Please clarify this.

To what extent are the results obtained here generalizable?

The first obvious limitations to the approach developed here is its application to competition networks, which is mentioned by the authors. But I believe that the limitations are stronger than that, I think the approach here would only work under certain assumptions and for specific datasets.

At the core of this comment there is a fundamental issue which is the definition of co-occurrence. A co-occurrence between two species is broadly defined as the presence of the two in the same spatial unit during an observation (so two species at the 'same somewhere' at the 'same sometime'). Assuming we consider only the kind of interactions considered here, I agree that 'species need to co-occur to interact'. But, to explore the relationship between the two kind of networks (co-occurrence vs biotic interaction), there is certainly a matter of scale. If the description of the co-occurrence were at a very fine scale, one could imagine a situation where there would be equivalence between the two networks (the mapping between the two network would be the identity function). Now at the other extreme of the spectrum, at very coarse spatial scale, most of the species would co-occur and so the pruning model would be different, with intense pruning for specialist species. In both cases, I wonder whether this pruning model described here would hold. In my opinion there is a matter of scale that at least need to be discussed.

A lack of null model?

One of the most important aspect of the classical debate about the relationship between co-occurrence and biotic interactions was the discussion about the null models, more precisely, the lack of null models in the original work of Diamond. Null models are crucial to determine whether a given pattern is expected under a given set of assumptions. As one stumbles into a new remarkable pattern, I think the first step is to ask whether the pattern is expected under a very minimal set of assumptions. In other words, the initial step should be to investigate whether the observed pattern departs from the simplest null model. In my opinion, in all the results presented here, there are no clear attempts to find out if the mapping between the two degree distributions (exponential -> scale-free) is expected under a simple model. According to the authors (page 3):

> *For instance, if interaction networks would result from a random removal of co-occurrence links, this would suggest that interactions are mostly contingent, and that the interaction network structure is trivially subordinated to the co-occurrence network. On the other hand, a systematic pruning can qualitatively alter the structure of the

7interaction network. For instance, if the pruning of the co-occurrence network makes the degree distribution change from exponential to a power-law, this would mean that co-occurrence patterns alone could be the result of chance conditioned by intraspecific environmental tolerance and dispersal abilities.*

Unless I missed something, there is no evidence that a random removal of links in the co-occurrence networks would provide different results. Actually my first guess was that a random removal of links should have yield similar results. I decided to do more than conjecturing: I carried out some numerical experiments (R code to reproduce it is attached to my review). I first created 1000 networks that would have a similar distribution to the one presented (I had to do a sum of two exponential distributions, but it worked), then I randomly removed links in the networks, I used three different "link removal intensities" (low, medium, high). See the results below.

In the figure presented, the dashed red lines represent the degree distribution for the co-occurrence networks and the dashed lines represent the networks obtained based on random removal, from left to the right the "link removal intensities" are low, medium and high. In my opinion, the results obtained under the high "link removal intensity" (rightmost panel) appear remarkably close to your results, which would mean that the co-occurrence networks in your case are biotic interaction networks, plus some random noise; some species co-occur with other with no particular reason. To be fair, your results might differ significantly from this very simple link removal model, but this needs to be demonstrated.

Another thought I have is that actually the simple linear pruning model you proposed could also be seen as a good null model. In essence, it basically means that a constant proportion of co-occurrence is removed for all species. If I am correct, I would say that is a very good null model, but I think it is hard to see how much ecology there is behind such model, and given that your results are well explained by this model, I find hard to understand the ecological discussion.

Taken my two first points here, I agree with the following statement

> *Our findings thus raise the critical question: How is an exponential degree distribution for the co-occurrence network converted into a power law when biotic interactions are considered?*

But I strongly disagree with the authors conclusions, based on the results, mine are as follows : 'the co-occurrence network observed here are biotic interaction networks plus some random noise, and `p` quantifies how much noise there is in the observed co-occurrence network'. So far I think that the results presented here confirmed that co-occurrence is often hard to interpret, it may hold an ecological signal, but this signal is blurry, at best.

Finally, assuming the pruning model employed here is ecologically meaningful and works well, several important questions remain:

- Can we infer biotic interaction networks from co-occurrence networks with this model?

- How can one obtain p from a co-occurrence network? (and what does p mean ecologically)
- What would be the quality of the predictions made by such model? (I believe there would be many false negatives and many false positives)

It would be hard to obtain good answers to these questions, but they must be discussed.

Technical comments

Theoretical model

The authors provide a short theoretical investigation in the supplementary material but, I think the authors should put in more effort to better connect it with the main text. So far, the reader has to make educated guess about why this actually matters. To address this, I think we need a few sentences to explain how these results were actually used.

Also:

- In the first section the predator/consumer are only referred to as 'nodes', I think this needs to be clarified (e.g. a 'node' here is a consumer species).

- In the 'Linear pruning model' section, the authors force all predator/consumer species to have at least one prey. So why don't they shift the Binomial distribution accordingly, basically decreasing d^c_{α} by one and add one to the final results (I may have missed something here).

Code and reproducibility

As an increasing number of manuscripts analyze existing datasets to test emerging theories or pinpoint remarkable patterns, reproducibility has become increasingly more important. As a consequence, sharing data and code has become the norm rather than the exception (and best practices exist, some provided by journals, some in articles already published). I value the authors efforts in tidying, commenting and sharing their code, I however have a few concerns with the current supplementary material `Code_and_data_cooccurrence`.

First, there are a lot of lines of code. Out of curiosity, I used `[cloc](https://github.com/AIDanial/cloc)` and it turns out there are 4395 lines of code. Here, I believe the code is relatively redundant and could have been formatted otherwise. That being said, I do not really see the importance of reformatting the code as long as it does what it is supposed to do which is increasingly harder to determine as the number of lines increases. This brings me to my main point which is that it is relatively difficult to assess the quality of the code in its current form. First, line 216 of

`Code_and_data_cooccurrence/data-analysis_occurrence.R` there is a `output_garraf_pp_real` that does not exist anywhere else (I guess it should be either `output_garraf_pp_real_pred` or `output_garraf_pp_real_pre`). Second, given the number of lines of code, it would help to structure

9even more the code, e.g. separate data wrangling from plotting data. It would also be useful to add a "main" function that would execute all the analysis. Third, as all datasets have their own format, it would be useful to provide a short description of the datasets format in the document (i.e. adding more details to the README about this).

Minor comments

- **Figure 1**: I wonder if it would be worth adding the mass functions in additional panels as I think it may help readers to grasp the ecological implications of the two distributions. Also, I would slightly change the label of the y-axis of the existing panels to describe the probability it represents, I would add something like $P(\text{consumer species degree} \geq x)$.

- **Methods**:

> This set of probabilities p defines a process by which to randomly generate bipartite networks.

I think there is something wrong with the wording here.

- Table S1. I would add the number of sites for each datasets.

Author Rebuttal to Initial comments

Reviewer #1 (Remarks to the Author):

There is much to like about this paper, it is interesting, well written, and the analyses seem fascinating, as far as they go. I really want to believe it. Unfortunately, I must say that in its current form, I just don't yet believe the main premise of the study that co-occurrence is a necessary condition for interaction. In principle, I see the point. Especially things like mutualisms. However, for predator-prey, I'm less convinced. To some degree it depends on the quality of the data for the networks. And this is where I have some doubts. The current paper refers primarily to a previous NEE paper in order to find the 'details' of the networks studied. I was actually a reviewer of that previous paper as well, both for Nature and NEE. I liked it, but in that case, I also found it hard to really understand in depth the details of the datasets and food webs. Of course, it is hard to describe completely with so many datasets and so little space, but in this case, there is some important information being lost that I would need to 'believe' the proof of concept.

10We appreciate the positive feedback provided by the reviewer and the time taken to explain in detail their concerns. Below we provide detailed responses to each comment and we explain how we have addressed the issues in the new version of the manuscript.

Here, my specific objection refers to the statement that “co-occurrence is a necessary condition for interaction” and thus that “interaction networks are a subset of co-occurrence networks”. I like the idea, but in practice, I’m having trouble believing this, depending on the quality and resolution of the data (which I believe in some cases, are very coarse). The foremost example in my mind is behavioral avoidance by prey and/or local extinction caused by predators. In freshwater systems, for example, whole suites of species, including insects, amphibians and small fish never co-occur in the same pond/lake with larger fish. They are almost never found together. But this is the result of a strong interaction and a generalist predator, together with habitat heterogeneity in the distribution of predators. Same with large mammal herbivores, seed predators, and many others. Sometimes species are never found together because the interaction is too strong and the distributions are otherwise heterogeneous. These might be extreme, but I’m certain there are many other similar examples in the kinds of data used here. Anytime there are very strong interactions on some, but not all, species in the community; habitat heterogeneity in the distribution of the predator; generalist predators; I would think similar results could emerge. Even in the mutualism data, can we be certain that the link is not possible, or just not observed because the pollinator has preferences in particular locations (nesting sites, other needs) but would certainly interact with a species if circumstances allowed. I guess in both cases there is a difference between possible interactions and realized interactions, but I’m not sure these can be distinguished with the sorts of data that are compiled in these networks.

We have included a new supplementary text with the details of each dataset. We haven’t included it before to avoid repetition with the already published paper, but we realise that it is important to emphasise that the datasets used here are not exactly the same as the ones used in the paper published in NEE in 2022. Here we have only included the datasets for which the premise ‘co-occurrence is a necessary condition for interaction’ holds true: plant-pollinator and host-parasitoid interactions. Moreover, the datasets used here are the ones where biotic interactions were observed empirically at high resolution (details about sampling procedure can now be found in the supplementary material). All datasets for which interactions were inferred (instead of observed) were excluded from this analysis. Thus, although we are aware that there are always limitations when using empirical data, we believe that these datasets are among the best quality-wise when addressing empirically-sampled biotic interactions across multiple sites in a replicated manner.

There's almost no information given about how the co-occurrence networks were estimated; at what scale; with what resolution; what effort? All of these things critically influence a species co-occurrence (and likelihood of interaction). Similarly, how were interaction networks estimated?

We have improved the explanation of the 'Network construction' section of the methods. It was indeed not clear enough. Both the networks of co-occurrence and the networks of biotic interactions were empirically sampled (instead of estimated) by the authors of each dataset.

We now specify that both networks were constructed at the same spatial scale: 'It is important to notice that the spatial scale considered for the biotic interactions is the same as the one considered for co-occurrences. That is, presence or absence of interactions and co-occurrences between species was empirically observed in each spatial unit of each dataset. The dimensions of the spatial units of each dataset were chosen by the authors of the original papers describing these datasets to ensure the correct description of the network of biotic interactions. Therefore, if two species were observed together but not interacting we can be confident that is not a false negative. The use of the same spatial scale for co-occurrences and interactions allows us to directly compare the two.'

Moreover we have added in the discussion the importance of the spatial scale under consideration: 'However, it is important to acknowledge the influence of spatial scale when predicting biotic interaction networks from co-occurrences. The spatial scale at which species co-occurrences are assessed can impact the comparison between these two types of networks. Generally, coarser scales may require stronger pruning of the co-occurrence network to approximate the interaction network. In this study, we used the same spatial scale to assess co-occurrences and characterise biotic interactions, allowing for direct comparisons. Nonetheless, future research should delve into how this comparison changes with increasing spatial scale and whether there exists a threshold beyond which co-occurrences no longer serve as accurate predictors of biotic interaction networks.'

In short, Despite my serious questions and concerns about the premise and empirical estimates of the networks, there is much to like about this paper. It is well written, and given the data, the analyses seem well done. I found the explanations of the network metrics (degree distributions) and the meaning of these relationships between the two kinds of networks compelling. That said, for the following reasons, I

12just don't yet believe it.

-We have tried for decades to get process from pattern; it's almost never possible. At the least, I would need to see more simulation models indicating it's possibility, and more exploration of null expectations and scenarios

We have added a new conceptual figure to explain the mechanisms leading to our findings and different potential scenarios (see Figure 3 in text and in the response to reviewer 2). Additionally, we also include a new supplementary text that provides the mathematical demonstration of the shift in the degree distribution (Text S2).

We have also included a new null model to test whether the patterns observed could be obtained by a random pruning of the co-occurrence network. As we explain there, under such a scenario the degree distribution would not change from an exponential to an approximate power law. It would remain exponential, only increasing the exponential decay. This is not at all what is seen in empirical data, which show a decelerating decay of the probability distribution, to the point where it becomes indistinguishable from a power law. This behaviour implies that the pruning is not random, but instead disproportionately favours generalist species. The new simulations confirm that the patterns observed cannot be achieved by a random selection of links from the co-occurrence network, but instead there needs to be a systematic pruning to observe the shift in the shape of the degree distributions. Both the new conceptual figure and the new analyses (and the explanations included in the main text) help to illustrate and clarify this very important point.

-The datasets used are not well described, but it's not at all clear how the statement "interaction networks are a subset of co-occurrence networks" can be true in many of these datasets. I would need much more proof before I would believe the results. It's just not clear

As explained above, we have added more details about the datasets in the supplementary material. We think that for the datasets used in this manuscript the statement holds true given that we only use plant-pollinator interactions and host-parasitoid interactions, which both need the co-occurrence to actually interact. We agree with the reviewer that for predator-prey interactions or competitive interactions this statement is not necessarily true as we acknowledge in the manuscript, and further research is needed to extend the theory proposed here to other types of interactions. We now

13emphasize in the introduction that we only use data for which the premise holds true and we highlight in the discussion that further research is needed to develop this framework beyond this type of networks. We have also included in the figures a cartoon of the types of interactions analysed to make it more explicit.

Reviewer #2 (Remarks to the Author):

Comments to the Author

This is an interesting study in which authors present and discuss whether interactions between species can be inferred from co-occurrence data. Specifically, they compare networks of co-occurrence with networks of interactions. First, they found that 20% of co-occurrences correspond to interactions. Secondly, they showed that the degree distribution shifts from exponential in co-occurrence networks to scale-free in interaction networks. In general terms, I think the study is well written, and the main problem is clearly presented. My main concern is about the presentation of the mechanisms involved in the results.

We thank the reviewer for the feedback provided. We believe that the issues raised helped us to generate an improved version of the manuscript.

1. The hypothesis of the study should be more explicitly developed. Authors said that the pruning of the co-occurrence network makes the degree distribution change from exponential to a power-law. The co-occurrence patterns could be the result of intraspecific environmental tolerance and dispersal abilities, while the interaction network could be explained by the interplay between the biotic and abiotic niches of a species. In this context, I can't see the mechanistic understanding of the shift in the degree distribution between co-occurrence networks and interaction networks. Maybe a conceptual figure that explains how the mechanisms lead the change could be great. Additionally, how these mechanisms are incorporated in the models.

Following the reviewer suggestion we have included a new conceptual figure (see below) to explain the mechanism involved in the shift of the degree distribution from co-occurrences to biotic interactions. We have also modified the introduction to better explain the hypothesis and the objective of the study. We argue that the comparison between co-occurrence networks and biotic interaction networks offers a unique opportunity to investigate the interplay between the abiotic niche of a species (that sets the

14limits to its spatial distribution) and its biotic niche (its interacting partners, such as its prey or predators), and that the understanding of such interplay can elucidate key mechanisms of community assembly. For instance, if we observe a change in the degree distribution from exponential in the co-occurrence network to a power-law in the biotic interaction network, we could interpret that co-occurrence patterns are the result of chance conditioned by interspecific environmental tolerance and dispersal abilities; but the patterns of the actual interactions showcasing network organisation hint at a systematic bias in the pruning of links that benefits generalist species. We have also included a new supplementary text (Text S2) that provides the mathematical demonstration of this mechanism.Figure 4. The role of super-generalist species in the emergence of power-law degree distributions. The plot at the top of the figure illustrates different possibilities for the relationship between the number of potential interactions and the number of realised links in the network of biotic interactions. The yellow line shows the pattern observed in the data analysed in this study where generalist species in terms of potential interactions realise a disproportionate large number of those links in the network of biotic interactions. The non-linearity of this relationship makes the non-proportionality across species explicit.

This relationship between potential interactions and realised (biotic) interactions highlights the strong generalism of the species both in terms of their abiotic niche (i.e. larger occupancy in space and thus larger co-occurrence with other species) and their biotic niche (i.e. more biotic interactions among those co-occurring with them). We call these species super-generalists. The grey and green dashed lines represent two other possible cases: a constant proportion of realised links across species and the case where specialist species would realise a larger proportion of the potential links. The plot at the bottom of the figure illustrates the consequences of these patterns for the shift in the network degree distribution from co-occurrence to biotic interactions. Given that super-generalist species keep a larger proportion of their potential links than specialist species, the degree distribution changes from an exponential in the co-occurrence network to a power-law in the network of biotic interactions (yellow line), where the probability of finding a species in the network with a large number of links is higher than in the other cases (grey and green dashed lines).

2. What about the sign of co-occurrence and interaction networks, and the mechanisms involved in the shift of the degree distribution.

We are not sure about what the reviewer means with this point. We did not observe a difference in the patterns between the positive interactions (plant-pollinator) and the negative interactions (host-parasitoid). In terms of co-occurrences, we only included interactions for which the co-occurrence is a necessary condition for the existence of the biotic interactions. That is, we did not include other types of interactions such as competition. The sign of the interaction does not play a role in the mechanism proposed to explain the shift of the degree distribution. The mechanism proposed is based on the interplay between species' abiotic niche and their biotic niche. We hope that with the new figure and its caption, this is now clear.

3. I did not understand why the 'super-generalists' pattern could explain the change from an exponential to a power-law degree distribution. Please, incorporate that in the conceptual figure, or a new one showing the role of the 'super-generalists' species, and how or why the frequency of co-occurrences are key predictors of degree distributions.

We did incorporate the role of super-generalists into generating the shift in the degree distribution in the new figure. We now show in the figure how the patterns would not emerge if the proportion of

17realised links would be constant across species. The reasoning behind the role of super-generalist species is that species that have a wide abiotic niche are able to be present in more sites and, therefore, co-occur with more potential interacting partners. But at the same time, they realise a disproportionately large amount of those potential links (i.e. larger proportion than the other species), showing that they also have a wide biotic niche. The fact that they do interact with more species allows the degree distribution to shift to a power-law in the biotic interaction network, because they promote the emergence of the characteristic fat-tail.

With our theoretical model we show that if we prune the co-occurrence network taking into account the frequency of interactions, we get patterns very similar to those observed in the empirical networks of interactions. This is because these super-generalist species not only co-occur with more species but they do so also with a higher frequency. Therefore, incorporating the frequency of co-occurrence in the model generates the disproportionality in the realised number of links that we observe in the empirical networks. We have now incorporated a new null model where the pruning of the co-occurrence network is made at random (i.e. it realises a constant proportion of interactions across species) and we show that the patterns that emerge from such a model do not resemble the empirical patterns. Thus, it further confirms the importance of this disproportionality in the number of realised links that is achieved through the role of the super-generalist species. Hopefully, all this is much more clear in this new version of the manuscript.

4. In the methods author said that: "...the expected number of links, L , conditioned on the fact that all consumers have at least one resource". This means that the richness or the size of the co-occurrence network and the interaction network should be the same. What are the implications of that?

Yes, co-occurrence networks and interaction networks involve the same number of species in the real data. The only thing that changes is the number of interactions involved, but all species are present in both networks. We have improved the explanation about network construction, which hopefully clarifies this point.

In our theoretical approach, we did not want to lose species when pruning the network of co-occurrences (to mimic the real data), and we did not want to have any isolated species in the interaction network (i.e. species present but without interactions), because that is also not the case in the empirical data. That is why we imposed the condition.

Reviewer #3 (Remarks to the Author):

Note that I added I also provide a pdf version of my review.

General comments

In this manuscript, the authors analyze 10 bipartite networks to scrutinize the relationship between the degree distribution of species in co-occurrence networks (also viewed here as networks of potential interactions) and the degree distribution of the corresponding networks of biotic interactions. Their results demonstrate that while the degree distribution in interaction networks is scale-free, the degree distribution in co-occurrence networks is exponential. Based on this, they propose a simple pruning model 'to approximate the realized network of biotic interactions' based on the co-occurrence network. The simple model they apply explains remarkably well how the exponential distribution of species degree observed in co-occurrence networks mapped into the scale-free degree distribution obtained in networks of biotic interactions. Based on this, they highlight the role of super-generalist species in structuring community.

The work achieved here is based on recent methodological developments and analyze high quality datasets to propose new insights on how co-occurrence networks structure inform the structure of biotic interaction networks. While I value the authors efforts, I believe major clarifications are required to better justify the importance of their findings. In what follows I summarize my main concerns around three major comments before providing further technical comments.

We thank the reviewer for his deep and constructive revision of our work.

Overall lack of clarity in methods

I am starting with this because the two following comments may partially be explained by this lack of clarity (hopefully, not just a lack of understanding on my end). In my opinion, the methods section should be clarified given that with the level of details provided so far, I seriously doubt that one can reproduce the analysis.

We have strongly modified the methods section to provide more clarity overall. Below we respond to each comment individually.

First, in 'Network construction' section:

> Therefore, a link between two species from different trophic levels is added when they co-occur in a given spatial unit of each dataset.

Does the spatial unit vary from one dataset to another? If so, does it matter (see the next comment)? Are the co-occurrence networks built here binary, meaning that a co-occurrence between two given species is set to '1' as soon as there is one co-occurrence event in at least one of the sites? If so, does that mean that there is no attempt to determine whether the co-occurrence is significant?

The spatial units do vary from dataset to dataset because their dimensions were decided by the authors of the original papers to properly characterise ecological networks in each system. However, we do not think it matters because the spatial unit used to characterise biotic interactions is the same as the one used to characterise co-occurrences across datasets. Our focus is to compare the network of biotic interactions with the network of co-occurrences within each dataset, therefore, we think that as long as the spatial unit is the same for each type of network analysed (co-occurrence and biotic interactions), the comparison is appropriate. We have included this argument in the manuscript in the 'Network construction' section.

The co-occurrence networks were indeed built as binary. However, we use the frequency of co-occurrence between species to develop our theoretical model. Our results show that considering the frequency of co-occurrence is fundamental to predict basic properties of the network of biotic interactions. We have specified in the 'Network construction' section that we use the frequency of co-occurrences to develop the theoretical model: 'In addition to the binary co-occurrence information (i.e. presence/absence of co-occurrence between species) used to build the network of co-occurrences, we also considered how frequently species from different trophic levels co-occur across sites to develop our theoretical model (see section below).'

Also,

20> Spatial co-occurrence is indeed used as a first step for inference of biotic interactions from auxiliary data.

I guess it is a general statement, not something you have done here, so this might actually bring some confusion. That said, it remains not clear how the biotic interactions were built here, I think the information is available for all the networks because they were built that way, is that correct? Please clarify this.

The sentence that the reviewer pointed out was indeed a general statement. We agree that it was adding confusion to how the co-occurrence networks were built, so we removed it from the text.

Regarding the biotic interactions, the reviewer is correct: all the information is available in the data. Biotic interaction networks were built based on empirically sampled interactions, and co-occurrence networks were built based on empirically-sampled presence/absence of species in each spatial unit. We have modified the 'Network construction' section of the methods and the caption of Figure 1 to clarify this point.

Second, in 'Network degree distribution' section:

> quantification of their degree distribution, defined as the probability $P(k)$

My understanding is that you have used the degree distribution of observed networks to obtain this distribution, so here one degree distribution is made of the collection of observed degrees in one network. Am I correct? Or are there replicates that I have missed (I would say no based on the data)?

You are correct. To characterise the degree distribution of both the network of biotic interactions and the co-occurrence networks we use the empirical data. Thus, there are no replicates for the degree distribution of the network of biotic interactions nor for the co-occurrence networks. There are replicates only for the pruning scenario given that it is a probabilistic model. We clarified this in the methods section.

Third, in 'Pruning of the co-occurrence network' section: a 'frequency of co-occurrence' is mentioned here, so the networks are no longer binary here? I am not sure I follow here.

As mentioned above, we have now included early on in the methods that we use the frequency of co-occurrences to develop the theoretical model. We have also improved the explanation of the model itself. The model uses the frequency of co-occurrences of two species across sites to determine their probability of interaction p_{ai} . Thus, the model generates an expected number of interactions for each species based on how frequently it co-occurs with the other species. With these expected number of interactions for each species, we can characterise the degree distribution of the pruned network. We hope this is now clear in the new description of the methods.

Last, It is quite hard to follow what models are used in figures 2-5 (and the corresponding supplementary figures) and some model details are missing.

> [...] we selected the most parsimonious function using the Akaike information criterion (AIC)

This pertains to the models used to generate the dashed line in Figure 2, right? What about the GAM used in Figure 4 (and S3), and why using GAM there instead of simpler models?

Yes, in Figure 2 we show the fit of the most parsimonious function for the degree distribution of the network of biotic interactions.

In Figure 4 (Figure 5 in the current version) we use GAM only for visualisation purposes given that we were not interested in characterise the specific shape of the relationship in a quantitative manner. We have specified in the figure caption that GAMs are only used for visualisation purposes.

In figures 3 and 5, my understanding is that you are using your pruning model, correct? Two question, how are the `p` for those models? In the code I have seen values of p

```
``R
# Garaf
p <- 0.07

# Nahuel
p <- 0.065

# Gottin
p <- 0.074
```

```
# Quercus  
p <- 0.0227
```

```
# Salix  
p <- 0.121  
'''
```

but I am not quite sure how they are obtained. Also, in figure 5, I do not understand how the different lines were obtained. It is mentioned that they are random realizations, but it is not clear to me what does it mean in this context. Please clarify this.

Yes, in Figure 3 and Figure 5 (which corresponds to Figure 4 and Figure 6, respectively in the current version) we show the predictions of our theoretical model. We have improved the explanation of the theoretical model in the methods section, including how do we obtain p and the reasoning behind the different randomisations of the model. We have 100 different random realisations because it is a probabilistic model. $p_{\alpha i}$ is the probability of interaction between consumer α and resource i , based on the frequency of co-occurrence between both species across sites $N_{\alpha i}$. Thus, any random realisation of the model defines the expected number of links for each species, which allows us to characterise the degree distribution and compare it to the actual degree distribution of the empirical network of biotic interactions. We hope this is clear in the new version.

Additionally, we have added a new supplementary table (Table S2) that provides the p for each dataset, and a new figure that illustrates the relationship between p and f (which is the proportion of realised interactions).

To what extent are the results obtained here generalizable?

The first obvious limitations to the approach developed here is its application to competition networks, which is mentioned by the authors. But I believe that the limitations are stronger than that, I think the approach here would only work under certain assumptions and for specific datasets.

At the core of this comment there is a fundamental issue which is the definition of co-occurrence. A co-occurrence between two species is broadly defined as the presence of the two in the same spatial unit during an observation (so two species at the 'same somewhere' at the 'same sometime'). Assuming we consider only the kind of interactions considered here, I agree that 'species need to co-occur to interact'.

23But, to explore the relationship between the two kind of networks (co-occurrence vs biotic interaction), there is certainly a matter of scale. If the description of the co-occurrence were at a very fine scale, one could imagine a situation where there would be equivalence between the two networks (the mapping between the two network would be the identity function). Now at the other extreme of the spectrum, at very coarse spatial scale, most of the species would co-occur and so the pruning model would be different, with intense pruning for specialist species. In both cases, I wonder whether this pruning model described here would hold. In my opinion there is a matter of scale that at least need to be discussed.

We agree with the reviewer that the spatial scale under consideration is an important factor to consider. As mentioned above, we believe that to make a meaningful comparison between co-occurrence and biotic interaction networks the spatial scale under consideration needs to guarantee that is fine enough to properly characterise biotic interactions. In this way, when observing co-occurrences between species that do not interact we can be confident that it is a true negative. Given that the datasets used here were specifically designed to characterise biotic interactions we are confident about the interpretation of our results. Moreover, the fact that the patterns observed are very similar across datasets (even though the spatial scale considered differs) and well-explained by our theoretical model, suggests that the mechanisms proposed are robust.

However, we agree that this is an important point that the reader needs to consider and that it was missing from our discussion. We have added in the discussion: 'it is important to acknowledge the influence of spatial scale when predicting biotic interaction networks from co-occurrences. The spatial scale at which species co-occurrences are assessed can impact the comparison between these two types of networks. Generally, coarser scales may require stronger pruning of the co-occurrence network to approximate the interaction network. In this study, we used the same spatial scale to assess co-occurrences and characterise biotic interactions, allowing for direct comparisons. Nonetheless, future research should delve into how this comparison changes with increasing spatial scale and whether there exists a threshold beyond which co-occurrences no longer serve as accurate predictors of biotic interaction networks.' Our key finding is thus that to interact, species not only need to co-occur, but co-occur *frequently* at the *appropriate scale*. Here we rely on expert knowledge and field expertise for this appropriate scale. It would be interesting to explore a more agnostic approach that would *reveal* that scale: Starting from the largest scale, where there cannot be any frequency of co-occurrence larger than one, increase the spatial resolution. At each resolution, apply the pruning model based on frequency of co-occurrence at that scale, to then determine at which scale the model converges to its best performance. We did not include this analysis here, as we think it would add extra complexity to an already technical paper. However, we agree that further research should focus on understanding how

this comparison changes as the spatial scale under consideration increases and whether there exists a limit at which co-occurrences are no longer a good predictor for biotic interaction networks.

A lack of null model?

One of the most important aspect of the classical debate about the relationship between co-occurrence and biotic interactions was the discussion about the null models, more precisely, the lack of null models in the original work of Diamond. Null models are crucial to determine whether a given pattern is expected under a given set of assumptions. As one stumbles into a new remarkable pattern, I think the first step is to ask whether the pattern is expected under a very minimal set of assumptions. In other words, the initial step should be to investigate whether the observed pattern departs from the simplest null model. In my opinion, in all the results presented here, there are no clear attempts to find out if the mapping between the two degree distributions (exponential -> scale-free) is expected under a simple model. According to the authors (page 3):

> *For instance, if interaction networks would result from a random removal of co-occurrence links, this would suggest that interactions are mostly contingent, and that the interaction network structure is trivially subordinated to the co-occurrence network. On the other hand, a systematic pruning can qualitatively alter the structure of the interaction network. For instance, if the pruning of the co-occurrence network makes the degree distribution change from exponential to a power-law, this would mean that co-occurrence patterns alone could be the result of chance conditioned by intraspecific environmental tolerance and dispersal abilities.*

Unless I missed something, there is no evidence that a random removal of links in the co-occurrence networks would provide different results. Actually my first guess was that a random removal of links should have yield similar results. I decided to do more than conjecturing: I carried out some numerical experiments (R code to reproduce it is attached to my review). I first created 1000 networks that would have a similar distribution to the one presented (I had to do a sum of two exponential distributions, but it worked), then I randomly removed links in the networks, I used three different "link removal intensities" (low, medium, high). See the results below.

In the figure presented, the dashed red lines represent the degree distribution for the co-occurrence

25networks and the dashed lines represent the networks obtained based on random removal, from left to the right the "link removal intensities" are low, medium and high. In my opinion, the results obtained under the high "link removal intensity" (rightmost panel) appear remarkably close to your results, which would mean that the co-occurrence networks in your case are biotic interaction networks, plus some random noise; some species co-occur with other with no particular reason. To be fair, your results might differ significantly from this very simple link removal model, but this needs to be demonstrated.

We fully agree with the reviewer. We had done those tests but did not include them in the old version of the manuscript. We have now included a new null model that illustrates how a random removal of links from the co-occurrence network does not generate the same results than the ones observed with the 'interaction rate' model, that considers the frequency of co-occurrences between species to prune the co-occurrence network, nor with the empirical data. With this null model, instead of having the probability of interaction between a given pair of species modulated by their frequency of co-occurrence, all species from opposite trophic levels that co-occur have the same probability of interacting. In this way, the resulting network of biotic interactions is a random subset of the co-occurrence network. The shape of the degree distribution of the pruned networks with this random model remained similar to the one observed in the co-occurrence networks, as suggested in the new conceptual figure (Figure 3) and the new supplementary text (Text S2) that provides the mathematical demonstration. The null model is explained in the methods section and its results shown and discussed in the main text. We also provided a new supplementary figure with all the plots illustrating the results (Figure S7).

Another thought I have is that actually the simple linear pruning model you proposed could also be seen as a good null model. In essence, it basically means that a constant proportion of co-occurrence is removed for all species. If I am correct, I would say that is a very good null model, but I think it is hard to see how much ecology there is behind such model, and given that your results are well explained by this model, I find hard to understand the ecological discussion.

A random pruning, as you say, amounts to removing a constant proportion of co-occurrences from all species. Then the degree distribution would not change from an exponential to an approximate power law. It would remain exponential, showing an increased exponential decay. To understand this mathematically, suppose that interaction degrees d are proportional to co-occurrence degrees c , so $d = f \times c$. Then, in terms of degree distributions, we have that $P(d > x) = P(fc > x) = P(c > x/f)$.

26This shows that the degree distribution for d , is just a rescaled version of the one of c . An exponential remains an exponential, a power law remains a power law. It is true that random noise could yield some qualitative departures when considering the degree distribution of a realised network, but for large enough networks this stochastic effect would vanish. This is not at all what is seen in empirical data, which across datasets we systematically observe a decelerating decay of the probability distributions, to the point where they become indistinguishable from power laws. This behaviour implies that the pruning is not random, but instead disproportionately favours generalist species. A way to interpolate between various scenarios is to see interaction degrees d , as a function $d = F(c)$ of co-occurrence degrees c . For instance, $F(c) = K \times (e^{ac} - 1)/a$, where a is a parameter that interpolates between scenarios and K is adjusted so that the fraction of expected links kept, is fixed and equal to f as a varies. Note that $F(c) \approx K(c + ac^2/2 + \dots)$. So for small a , we recover the linear model. For positive a , generalists (large c) keep a disproportionately large number of links. For negative a it is the opposite. Then, $P(d > x) = P(F(c) > x) = P(c > F^{-1}(x))$. Because $F^{-1}(x) = \frac{1}{a} \log(1 + ax/K)$, if the degree distribution of cooccurrences is exponential, so $P(c > x) = e^{-Ax}$ then $P(d > x) = (1 + \frac{ax}{K})^{-A/a}$. This is an exponential for a close to 0 but it approaches a power law for positive a . For negative a , the probability $P(d > x)$ drops to zero at finite x . These different scenarios are represented in the new conceptual figure and explained in the new supplementary Text S2.

In the interaction-rate model that we propose, the proportion of links that is removed is not constant across species, it depends on the frequency of co-occurrence. This is a key point to understand the shift in the degree distribution from exponential in the co-occurrence networks to power-law in the biotic interaction networks. In Figure 4 we show this non-proportionality across species: generalist species in terms of potential interactions realise a disproportionate large amount of those links in the network of biotic interactions. The non-linearity of this relationship shows explicitly how the proportion of co-occurrences removed is not constant across species either in the empirical data analysed nor the theoretical model proposed. The new null model does take a constant proportion across species.

All this is now explained in the new Figure 3, in the main text and in a new supplementary text that provides a mathematical demonstration (Text S2).

Taken my two first points here, I agree with the following statement

> *Our findings thus raise the critical question: How is an exponential degree distribution for the co-occurrence network converted into a power law when biotic interactions are considered?*

But I strongly disagree with the authors conclusions, based on the results, mine are as follows : 'the co-occurrence network observed here are biotic interaction networks plus some random noise, and p quantifies how much noise there is in the observed co-occurrence network'. So far I think that the results presented here confirmed that co-occurrence is often hard to interpret, it may hold an ecological signal, but this signal is blurry, at best.

We hope that with the new conceptual figure and the results of the null model, the mechanism that we propose is now clear and convincing. Our results show that the role of super-generalist species is fundamental to observe the shift in the degree distribution. We show with the new null model that the observed patterns would not emerge if the proportion of realised links would be constant across species. In the empirical data, super-generalist species have a wider abiotic niche that allow them to be present in more sites and, therefore, co-occur with more potential interacting partners. But at the same time, they realise a disproportionately large amount of those potential links (i.e. larger proportion than the other species), showing that they also have a wider biotic niche. The fact that they do interact with more species allows the degree distribution to shift to a power-law in the biotic interaction network, because they promote the emergence of the characteristic fat-tail. With our theoretical model we show that if we prune the co-occurrence network taking into account the frequency of interactions, we get patterns very similar to those observed in the empirical networks of interactions. This is because these super-generalist species not only co-occur with more species but they do so also with a higher frequency. Therefore, incorporating the frequency of co-occurrence in the model generates the disproportionality in the realised number of links that we observe in the empirical networks.

Instead, when comparing the patterns observed in the empirical data with the results of the null model we can see that there is no shift in the shape of the degree distribution. There needs to be a systematic bias in the pruning of links that benefits generalist species to shift the degree distribution from exponential to power-law. Thus, we do not think that co-occurrence networks only contain the information about biotic interactions plus noise. By comparing co-occurrence networks to biotic interaction networks we can elucidate key mechanisms of community assembly such as the interplay between species biotic and abiotic niche.

We discuss about p in the following comment.

Finally, assuming the pruning model employed here is ecologically meaningful and works well, several important questions remain:

- Can we infer biotic interaction networks from co-occurrence networks with this model?
- How can one obtain p from a co-occurrence network? (and what does p mean ecologically)
- What would be the quality of the predictions made by such model? (I believe there would be many false negatives and many false positives)

It would be hard to obtain good answers to these questions, but they must be discussed.

These are very good points that were not covered in the previous version of the manuscript.

Indeed, we need p to be able to make predictions about the network of biotic interactions from co-occurrences. We now better explain, both in methods and in the results and discussion, how we estimate p and what are the values obtained across datasets. p is the per-site interaction rate, and we estimate it by knowing how many links need to be pruned from the co-occurrence network. Similar to the proportion of realised links, we found a strong generality across datasets (mean = 0.077, sd = 0.028). Thus, we propose that the generality found here can be used as a reference value to generate predictions of biotic interaction networks without further information than co-occurrence data. Moreover, we illustrate in a new supplementary figure (Figure S6), that p is well correlated with the proportion of co-occurrence interactions that are actual biotic interactions (f). Given that the per-site interaction rate is likely easier to estimate in empirical data than the full network of biotic interactions, we suggest that this relationship can provide valuable information.

Throughout the text we make explicit that what the model is able to predict is basic properties of the network of biotic interactions, such as the degree distribution, but not the specific interactions. For that further research is needed, as we state in the conclusion.

Technical comments

29Theoretical model

The authors provide a short theoretical investigation in the supplementary material but, I think the authors should put in more effort to better connect it with the main text. So far, the reader has to make educated guess about why this actually matters. To address this, I think we need a few sentences to explain how these results were actually used.

You were right, that part of the theory was not used in the manuscript. It was a generalisation of the model, but given that it was not used in the analyses presented here we decided to remove it from the supplementary material because we believe it could bring unnecessary confusion.

Also:

- In the first section the predator/consumer are only referred to as 'nodes', I think this needs to be clarified (e.g. a 'node' here is a consumer species).

Not only, we also analyse the patterns for resource species and such results are shown in the supplementary material.

- In the 'Linear pruning model' section, the authors force all predator/consumer species to have at least one prey. So why don't they shift the Binomial distribution accordingly, basically decreasing α by one and add one to the final results (I may have missed something here).

If we understand correctly, It seems that you would suggest (i) removing one link per predator species, (ii) prune the network without forcing predators to have at least one prey (iii) finally add one link per predator. This could probably work. In our simulations we use a different algorithm, we sequentially prune the links from each predator but repeat the pruning for that species if it is left with no prey, before moving on to the next predator. Importantly, how we enforce this condition does not change the analytical expression for the conditional total expected number of links, which is a function of our parameter p . We then use this function to choose p such that this expected value matches the empirical data.

Code and reproducibility

30As an increasing number of manuscripts analyze existing datasets to test emerging theories or pinpoint remarkable patterns, reproducibility has become increasingly more important. As a consequence, sharing data and code has become the norm rather than the exception (and best practices exist, some provided by journals, some in articles already published). I value the authors efforts in tidying, commenting and sharing their code, I however have a few concerns with the current supplementary material ``Code_and_data_cooccurrence``.

First, there are a lot of lines of code. Out of curiosity, I used [cloc](<https://github.com/AlDanial/cloc>) and it turns out there are 4395 lines of code. Here, I believe the code is relatively redundant and could have been formatted otherwise. That being said, I do not really see the importance of reformatting the code as long as it does what it is supposed to do which is increasingly harder to determine as the number of lines increases. This brings me to my main point which is that it is relatively difficult to assess the quality of the code in its current form. First, line 216 of ``Code_and_data_cooccurrence/data-analysis_occurrence.R`` there is a ``output_garraff_pp_real`` that does not exist anywhere else (I guess it should be either ``output_garraff_pp_real_pred`` or ``output_garraff_pp_real_prey``). Second, given the number of lines of code, it would help to structure even more the code, e.g. separate data wrangling from plotting data. It would also be useful to add a "main" function that would execute all the analysis. Third, as all datasets have their own format, it would be useful to provide a short description of the datasets format in the document (i.e. adding more details to the README about this).

Thank you for taking the time to check all this and make useful suggestions to improve the code as well.

We have improved the structure of the code by removing all the unnecessary lines in the `data_processing_occurrence.R` files and structuring the `data_analyses_occurrence.R` file. The `data_analyses` file is now structured following the order of the figures in the manuscript, so it is much more clear how to get each of the outputs shown in the manuscript. Moreover, we have added more comments to make it easier to follow. We did not include a main function that executes all the analyses because there are several output files that are executed in different parts of the code and that can be later used for different purposes. Depending on their interests, the users might want to only execute specific parts of the analyses. With the new comments and structure we hope the code is now more friendly and reproducible. We have also added in the README more details about the format of each dataset.

Minor comments

- **Figure 1**: I wonder if it would be worth adding the mass functions in additional panels as I think it may help readers to grasp the ecological implications of the two distributions. Also, I would slightly change the label of the y-axis of the existing panels to describe the probability it represents, I would add something like $P(\text{consumer species degree} \geq x)$.

We have changed the y-axis according to your suggestion. We did not add new panels with the mass functions because we already have a few large figures and because we thought it could bring some confusion given that the rest of the results are presented as cumulative distributions.

- **Methods**:

> This set of probabilities p defines a process by which to randomly generate bipartite networks.

I think there is something wrong with the wording here.

We revised this section.

- Table S1. I would add the number of sites for each datasets.

Thanks for the suggestion. We have added the number of sites for each dataset.

Kevin Cazelles

Decision Letter, first revision:6th September 2023

Dear Dr. Galiana,

Thank you for submitting your revised manuscript "From species co-occurrences to interactions: the emergence of power-law degree distributions" (NATECOLEVOL-23051045A). It has now been seen again by the original reviewers and their comments are below. The reviewers find that the paper has improved in revision, and therefore we'll be happy in principle to publish it in Nature Ecology & Evolution, pending minor revisions to satisfy the reviewers' final requests and to comply with our editorial and formatting guidelines.

[REDACTED]

Reviewer #1 (Remarks to the Author):

The authors have done a remarkable job at addressing my comments, as well as those of the other reviewers. I find the new conceptual figure and null model analyses, as well as the descriptions of the datasets and their adherence to assumptions appropriate. I have no further comments.

Reviewer #2 (Remarks to the Author):

I have read the authors responses and the revision of this manuscript, and I think that, the authors have adequately addressed all my concerns. On the whole, as it is the ms is much improved and clearer presented.

I think this study will contribute to advance in the connection between co-occurrence patterns and biotic interaction networks, providing valuable information about the drivers of community assembly rules.

Reviewer #3 (Remarks to the Author):

I believe the revised manuscript is much more clear, I am pleased that the authors took all my

33criticisms seriously and use them to improve their work (e.g. the addition of the NULL models in SI and the new section 'Beyond random pruning of co-occurrences'). I still wonder whether actual predictions could be made, but can be considered as beyond the scope of this study.

At this stage I believe I did my job as a reviewer providing a constructive review of the authors' work, and the authors did theirs by answering all my concerns providing all the necessary details as well as additional content/analyses where needed. The result of this is a work that will be a good contribution to the literature (in my humble opinion).

Minor comments

2 minor comments in Text S2:

- $\$F\$$ may not be an increasing function (but I think this is just an assumption)
- I think it is worth mentioning that $\$E\$$ denotes the expected value.

Our ref: NATECOLEVOL-23051045A

25th September 2023

Dear Dr. Arnoldi,

Thank you for your patience as we've prepared the guidelines for final submission of your Nature Ecology & Evolution manuscript, "From species co-occurrences to interactions: the emergence of power-law degree distributions" (NATECOLEVOL-23051045A). Please carefully follow the step-by-step instructions provided in the attached file, and add a response in each row of the table to indicate the changes that you have made. Please also check and comment on any additional marked-up edits we have proposed within the text. Ensuring that each point is addressed will help to ensure that your revised manuscript can be swiftly handed over to our production team.

We would like to start working on your revised paper, with all of the requested files and forms, as soon as possible (preferably within two weeks). Please get in contact with us immediately if you anticipate it taking more than two weeks to submit these revised files.

34When you upload your final materials, please include a point-by-point response to any remaining reviewer comments.

In recognition of the time and expertise our reviewers provide to Nature Ecology & Evolution's editorial process, we would like to formally acknowledge their contribution to the external peer review of your manuscript entitled "From species co-occurrences to interactions: the emergence of power-law degree distributions". For those reviewers who give their assent, we will be publishing their names alongside the published article.

Nature Ecology & Evolution offers a Transparent Peer Review option for new original research manuscripts submitted after December 1st, 2019. As part of this initiative, we encourage our authors to support increased transparency into the peer review process by agreeing to have the reviewer comments, author rebuttal letters, and editorial decision letters published as a Supplementary item. When you submit your final files please clearly state in your cover letter whether or not you would like to participate in this initiative. Please note that failure to state your preference will result in delays in accepting your manuscript for publication.

Cover suggestions

We welcome submissions of artwork for consideration for our cover. For more information, please see our https://www.nature.com/documents/Nature_covers_author_guide.pdf guide for cover artwork.

Nature Ecology & Evolution has now transitioned to a unified Rights Collection system which will allow our Author Services team to quickly and easily collect the rights and permissions required to publish your work. Approximately 10 days after your paper is formally accepted, you will receive an email in providing you with a link to complete the grant of rights. If your paper is eligible for Open Access, our Author Services team will also be in touch regarding any additional information that may be required to arrange payment for your article.

Please note that *Nature Ecology & Evolution* is a Transformative Journal (TJ). Authors may publish their research with us through the traditional subscription access route or make their paper

35immediately open access through payment of an article-processing charge (APC). Authors will not be required to make a final decision about access to their article until it has been accepted. [Find out more about Transformative Journals](https://www.springernature.com/gp/open-research/transformative-journals)

Authors may need to take specific actions to achieve [compliance with funder and institutional open access mandates](https://www.springernature.com/gp/open-research/funding/policy-compliance-faqs). If your research is supported by a funder that requires immediate open access (e.g. according to [Plan S principles](https://www.springernature.com/gp/open-research/plan-s-compliance)) then you should select the gold OA route, and we will direct you to the compliant route where possible. For authors selecting the subscription publication route, the journal's standard licensing terms will need to be accepted, including [self-archiving-and-license-to-publish](https://www.nature.com/nature-portfolio/editorial-policies/self-archiving-and-license-to-publish). Those licensing terms will supersede any other terms that the author or any third party may assert apply to any version of the manuscript.

[REDACTED]

[REDACTED]

Reviewer #1:

Remarks to the Author:

The authors have done a remarkable job at addressing my comments, as well as those of the other reviewers. I find the new conceptual figure and null model analyses, as well as the descriptions of the datasets and their adherence to assumptions appropriate. I have no further comments.

Reviewer #2:

Remarks to the Author:

I have read the authors responses and the revision of this manuscript, and I think that, the authors have adequately addressed all my concerns. On the whole, as it is the ms is much improved and clearer presented.

I think this study will contribute to advance in the connection between co-occurrence patterns and biotic interaction networks, providing valuable information about the drivers of community assembly rules.

Reviewer #3:

Remarks to the Author:

I believe the revised manuscript is much more clear, I am pleased that the authors took all my criticisms seriously and use them to improve their work (e.g. the addition of the NULL models in SI and the new section 'Beyond random pruning of co-occurrences'). I still wonder whether actual predictions could be made, but can be considered as beyond the scope of this study.

At this stage I believe I did my job as a reviewer providing a constructive review of the authors' work, and the authors did theirs by answering all my concerns providing all the necessary details as well as additional content/analyses where needed. The result of this is a work that will be a good contribution to the literature (in my humble option).

Minor comments

2 minor comments in Text S2:

- F may not be an increasing function (but I think this is just an assumption)
- I think it is worth mention that E denotes the expected value.

Final Decision Letter:

17th October 2023

Dear Dr Galiana,

We are pleased to inform you that your Article entitled "Power-laws in species' biotic interaction networks can be inferred from co-occurrence data", has now been accepted for publication in Nature Ecology & Evolution.

Over the next few weeks, your paper will be copyedited to ensure that it conforms to Nature Ecology and Evolution style. Once your paper is typeset, you will receive an email with a link to choose the appropriate publishing options for your paper and our Author Services team will be in touch regarding any additional information that may be required

After the grant of rights is completed, you will receive a link to your electronic proof via email with a request to make any corrections within 48 hours. If, when you receive your proof, you cannot meet

37this deadline, please inform us at rjsproduction@springernature.com immediately.

Due to the importance of these deadlines, we ask you please us know now whether you will be difficult to contact over the next month. If this is the case, we ask you provide us with the contact information (email, phone and fax) of someone who will be able to check the proofs on your behalf, and who will be available to address any last-minute problems . Once your paper has been scheduled for online publication, the Nature press office will be in touch to confirm the details.

Acceptance of your manuscript is conditional on all authors' agreement with our publication policies (see www.nature.com/authors/policies/index.html). In particular your manuscript must not be published elsewhere and there must be no announcement of the work to any media outlet until the publication date (the day on which it is uploaded onto our web site).

Please note that *Nature Ecology & Evolution* is a Transformative Journal (TJ). Authors may publish their research with us through the traditional subscription access route or make their paper immediately open access through payment of an article-processing charge (APC). Authors will not be required to make a final decision about access to their article until it has been accepted. [Find out more about Transformative Journals](https://www.springernature.com/gp/open-research/transformative-journals)

Authors may need to take specific actions to achieve [compliance](https://www.springernature.com/gp/open-research/funding/policy-compliance-faqs) with funder and institutional open access mandates. If your research is supported by a funder that requires immediate open access (e.g. according to [Plan S principles](https://www.springernature.com/gp/open-research/plan-s-compliance)) then you should select the gold OA route, and we will direct you to the compliant route where possible. For authors selecting the subscription publication route, the journal's standard licensing terms will need to be accepted, including [those licensing terms](https://www.nature.com/nature-portfolio/editorial-policies/self-archiving-and-license-to-publish) will supersede any other terms that the author or any third party may assert apply to any version of the manuscript.

We welcome the submission of potential cover material (including a short caption of around 40 words)

38related to your manuscript; suggestions should be sent to Nature Ecology & Evolution as electronic files (the image should be 300 dpi at 210 x 297 mm in either TIFF or JPEG format). Please note that such pictures should be selected more for their aesthetic appeal than for their scientific content, and that colour images work better than black and white or grayscale images. Please do not try to design a cover with the Nature Ecology & Evolution logo etc., and please do not submit composites of images related to your work. I am sure you will understand that we cannot make any promise as to whether any of your suggestions might be selected for the cover of the journal.

You can generate the link yourself when you receive your article DOI by entering it here: <http://authors.springernature.com/share>.

[REDACTED]

P.S. Click on the following link if you would like to recommend Nature Ecology & Evolution to your librarian <http://www.nature.com/subscriptions/recommend.html#forms>

** Visit the Springer Nature Editorial and Publishing website at http://editorial-jobs.springernature.com?utm_source=ejp_NEcoE_email&utm_medium=ejp_NEcoE_email&utm_campaign=ejp_NEcoE for more information about our career opportunities. If you have any questions please click [here](mailto:editorial.publishing.jobs@springernature.com).**